# Deep Clinical Phenotyping of Schizophrenia Spectrum Disorders Using Data-Driven Methods: Marching towards Precision Psychiatry

**DOI:** 10.3390/jpm13060954

**Published:** 2023-06-05

**Authors:** Tesfa Dejenie Habtewold, Jiasi Hao, Edith J. Liemburg, Nalan Baştürk, Richard Bruggeman, Behrooz Z. Alizadeh

**Affiliations:** 1Department of Epidemiology, University Medical Center Groningen, University of Groningen, 9700 RB Groningen, The Netherlands; j.hao@umcg.nl; 2Department of Psychiatry, University Medical Center Groningen, University Center for Psychiatry, Rob Giel Research Center, University of Groningen, 9700 RB Groningen, The Netherlands; e.j.liemburg@umcg.nl (E.J.L.); r.bruggeman@umcg.nl (R.B.); 3Department of Quantitative Economics, School of Business and Economics, Maastricht University, 6200 MD Maastricht, The Netherlands; n.basturk@maastrichtuniversity.nl; 4Department of Clinical and Developmental Neuropsychology, Faculty of Behavioural and Social Sciences, University of Groningen, 9700 RB Groningen, The Netherlands

**Keywords:** schizophrenia spectrum disorder, psychosis, phenotype, deep phenotyping, cognitive impairment, data-driven methods, precision psychiatry

## Abstract

Heterogeneity is the main challenge in the traditional classification of mental disorders, including schizophrenia spectrum disorders (SSD). This can be partly attributed to the absence of objective diagnostic criteria and the multidimensional nature of symptoms and their associated factors. This article provides an overview of findings from the Genetic Risk and Outcome of Psychosis (GROUP) cohort study on the deep clinical phenotyping of schizophrenia spectrum disorders targeting positive and negative symptoms, cognitive impairments and psychosocial functioning. Three to four latent subtypes of positive and negative symptoms were identified in patients, siblings and controls, whereas four to six latent cognitive subtypes were identified. Five latent subtypes of psychosocial function—multidimensional social inclusion and premorbid adjustment—were also identified in patients. We discovered that the identified subtypes had mixed profiles and exhibited stable, deteriorating, relapsing and ameliorating longitudinal courses over time. Baseline positive and negative symptoms, premorbid adjustment, psychotic-like experiences, health-related quality of life and PRS_SCZ_ were found to be the strong predictors of the identified subtypes. Our findings are comprehensive, novel and of clinical interest for precisely identifying high-risk population groups, patients with good or poor disease prognosis and the selection of optimal intervention, ultimately fostering precision psychiatry by tackling diagnostic and treatment selection challenges pertaining to heterogeneity.

## 1. Introduction

Schizophrenia is one of the most common and severe psychiatric disorders, with a lifetime prevalence of approximately 1% [1]. Globally, schizophrenia cases almost doubled from 13.1 million in 1990 to 20.9 million in 2016, and it has contributed to 13.4 million years of life lived with disability, which brings a substantial burden on patients, families, communities and healthcare systems [2].

The first episode of schizophrenia usually occurs in late adolescence or early adulthood [1]. The hallmark manifestations of schizophrenia include positive (i.e., delusions, hallucinations and disorganized behaviour) and negative (i.e., blunted affect and asociality) symptoms [3]. Positive symptoms are the main presenting complaints and reason that patients seek treatment [4]. They are characterized by relapses and remission, which are often managed through antipsychotic treatment [4]. Negative symptoms are more stable, persistent and difficult to treat using pharmacological and non-pharmacological interventions [5]. Negative symptoms are believed to often occur prior to the onset of schizophrenia [5].

Cognitive deficits are also common symptoms observed in up to 90% of patients with schizophrenia, although they are not yet included as a diagnostic criterion of schizophrenia in DSM [6,7]. The most affected cognitive functions are episodic and working memory, attention, verbal fluency, executive function, problem-solving, processing speed and social cognition [8]. Cognitive deficits are stable and considered heritable (h^2^ = 50%) traits of schizophrenia [9]. Similar to negative symptoms, cognitive symptoms could also occur before the onset of schizophrenia [5] and barely respond to antipsychotic treatment [10]. 

Positive, negative and cognitive symptoms have pervasive impacts on the psychosocial functioning, occupational and academic performance, and quality of life of a patient [5,7]. Furthermore, it is noteworthy to indicate that these phenotypes are also diversely present in siblings due to shared genetic and non-genetic factors with their probands [6]. 

The *Diagnostic and Statistical Manual of Mental Disorders* (DSM) is the most widespread nosographic manual used by international scientific and clinical communities to diagnose mental disorders [3,11]. Clinicians subjectively interview patients and diagnose based on the set of criteria listed in the DSM [11]. However, the DSM has been criticized for the inability of diagnostic categories to map cleanly onto either biology or outcome, due to its overreliance on positive symptoms and negligence of affective, negative and cognitive symptoms [12]. For example, this leads to up to a 50% misdiagnosis rate in schizophrenia [13]. The high rate of misdiagnosis and the consequent misclassifications add a considerable burden due to ineffective interventions, poor disease progression, resource wastage and inconsistency in research reports [13]. Moreover, attempts to identify objective diagnostic biomarkers, such as genetic, proteomic and immunological markers, for schizophrenia have not yet been successful because of methodological shortcomings, professional indecision and conceptual contradictions [14]. Lastly, both psychiatric and non-psychiatric comorbidities complicate the treatment and prediction of schizophrenia outcomes [15]. Therefore, the diagnostic methods require reconsideration for a better translation of evidence into accurate and consistent clinical practice.

Several solutions have been proposed for tackling the aforementioned issues. First, DSM-5 has attempted to shift away from categorical diagnostics and adopted a dimensional approach that emphasizes the underlying specific symptoms or symptom clusters and their relationship with each other [3]. DSM-5 introduced the concept of schizophrenia spectrum disorders (SSD), encompassing various diagnoses, including schizophrenia, although in practice, diagnoses remain exclusive rather than to be seen as spectrum of disorders [3]. Another thriving solution is the use of variable and patient-centred data-driven approaches that classify psychiatric disorders to find more homogeneous subgroups of individuals in the exploitation of multidimensional symptom profiles [16]. 

Up till now, the narratives have shown the inherent phenotypic heterogeneity of SSD, which consequently and urgently calls for deep clinical phenotyping using large-scale studies with the possibility of in-depth exploration of disease subtypes and characterization. Previous epidemiological studies on SSD were hampered by limited sample size, limited use of omics data, and short duration of follow-up [2]. Of interest, no study comprehensively examined phenotypes of SSD and underlying factors in patients, unaffected siblings and controls. 

Detecting phenotypic subgroups of patients suffering from complex diseases is an important element of precision medicine and could provide support for the early detection of deteriorating patients, prediction of individualised and customised treatment, and prevention strategies for different phenotypic groups, which ultimately results in enhanced treatment outcome [17,18]. The in-depth analysis of phenotype plays a key role in clinical practice and medical research [18]. This article provides an overview of findings from eight Genetic Risk and Outcome of Psychosis (GROUP) cohort studies on the deep clinical phenotyping of schizophrenia spectrum disorders targeting positive and negative symptoms [19,20], cognitive impairments [21,22,23] and psychosocial functioning [24,25,26]. Briefly, we present identified latent subtypes (i.e., subphenotypes), their longitudinal course or profiles of subtypes and important predictors. Finally, we present future research directions to extend this line of research. 

## 2. Methods and Materials

### 2.1. Study Population

Patients, siblings and controls participated in the Genetic Risk and Outcome of Psychosis (GROUP) cohort study. GROUP is a six-year, longitudinal cohort study in the Netherlands [27]. Patients were included if they were diagnosed with a schizophrenia spectrum disorder (53.8% schizophrenia), were aged between 16 and 50 years old, had a good command of the Dutch language and were willing and capable of giving written informed consent. Siblings and controls were included if they met the above criteria and had no known lifetime psychotic disorder. At baseline, 1119 patients, 1059 first-degree siblings and 586 healthy controls were recruited. Details on the GROUP project structure, recruitment of participants, sample size estimation, data collection and ethical approval have been published elsewhere [27]. 

### 2.2. Measurements

Data were collected at baseline and after three years and six years of follow-up. DSM-4 [28] was used to diagnose schizophrenia spectrum disorder. Positive and negative symptoms in patients were assessed at baseline and after three years and six years using the Positive and Negative Syndrome Scale (PANSS) [19,20]. Similarly, positive and negative schizotypy (i.e., subclinical symptoms) in siblings and healthy controls were assessed with the Structured Interview for Schizotypy-revised (SIS-R) [19,20]. Positive and negative subscales’ mean or sum score was used to quantify the severity of symptoms. Cognitive function was assessed in patients, siblings and controls at baseline and after three years and six years using the Measurement and Treatment Research to Improve Cognition in Schizophrenia (MATRICS) consensus cognitive battery test [21,22,23]. Specifically, we used the Word Learning Task (WLT) (i.e., to measure memory), Continuous Performance Test-HQ (CPT-HQ) (i.e., to measure executive function) and Wechsler Adult Intelligence Scale (WAIS-III) (i.e., to measure IQ). Subdomain and global cognitive function were evaluated by creating a composite score using the Z-score [22,23] or principal component analysis methods [21]. Social function was measured after three years and six years by using the Social Functioning Scale (SFS), whereas quality of life was measured at baseline and after three years and six years with the brief version of the World Health Organization Quality of Life assessment (WHOQOL-BREF) [24]. A multidimensional social inclusion composite score was created from three years measurement of the SFS and WHOQOL-BREF using the Z-score method [24]. Premorbid adjustment was retrospectively measured using the premorbid adjustment sale (PAS) in three life periods: childhood (<12 years), early adolescence (12–16 years) and late adolescence (16–19 years) [25,26]. All information was gathered from parents or siblings and a mean score was created to quantify overall premorbid functioning [25,26]. Sociodemographic, genetic and clinical data were also collected. DNA was extracted from peripheral blood lymphocytes in a 20 mL stored blood sample. Genotyping, quality control and imputation were carried out following the standard procedure [19,21]. The polygenic risk score for schizophrenia (PRS_SCZ_) was calculated using Psychiatric Genomics Consortium (PGC) schizophrenia genome-wide association study (GWAS) summary statistics, where a higher PRS_SCZ_ indicates a higher genetic vulnerability to developing schizophrenia [21]. 

### 2.3. Data Analyses

*Identification of latent subtypes*: Person-centred latent class models were used to identify subgroups of individuals with similar phenotypes, which we call latent subtypes throughout this article. In five studies [19,20,21,22,26], we applied group-based trajectory modelling (GBTM) to identify latent subtypes and evaluate their longitudinal course throughout the six-year follow-up, whereas we applied K-means clustering in three studies [23,24,25] to identify latent subtypes at one time point (e.g., baseline data, or three-year follow-up). The analyses were carried out using SAS and R software.

*Predictors of latent subtypes*: In three studies [19,21,22], mixed-effects regression models were applied to investigate predictors of latent subtypes. The other studies [20,23,24,25] used analysis of variance (ANOVA) or the Kruskal–Wallis test and chi-square test to investigate continuous and categorical variables that distinguished latent subtypes, respectively. The analyses were carried out using SAS and R software. 

## 3. Results

### 3.1. Latent Subtypes

*Positive and negative symptoms*: In patients, we identified three latent subtypes of positive and negative symptoms [19] (Figure 1A). Additionally, using social amotivation and expressive deficit subphenotypes of negative symptoms, we identified four latent subtypes in patients [20]. In siblings, we consistently distinguished four latent subtypes of positive and negative schizotypy (Figure 1B), while three positive and negative schizotypy latent subtypes were identified in controls (Figure 1C) [19].

*Cognitive functions*: We identified five and four cognitive subtypes in patients and their siblings, respectively (Figure 1A,B) [21,22]. In another study using cross-sectional data, we identified three cognitive subtypes in siblings [23]. In controls, we identified four cognitive subtypes (Figure 1C) [21]. After combing all samples together, we unravelled six latent cognitive subtypes (Figure 2) [21]. Furthermore, we examined the latent subtypes of three cognitive subdomains—verbal learning and memory, sustained attention and vigilance, and global cognitive function (IQ) [29]. In patients, we identified five latent subtypes of global cognitive function (IQ), and three subtypes of verbal learning and memory, and sustained attention and vigilance. In siblings and controls, five, four and three subtypes of global cognitive function (IQ), verbal learning and memory, and sustained attention and vigilance were identified, respectively. In all combined samples (Figure 2), we identified five subtypes of verbal learning and memory, and global cognitive function (IQ) domains, whereas four subtypes of sustained attention and vigilance were identified. 

*Psychosocial functions*: In patients, we identified five latent subtypes of multidimensional social inclusion [24]. In addition, we revealed six latent subtypes of premorbid adjustment in patients [25,26]. 

### 3.2. Longitudinal Courses and Profiles of Subtypes

*Positive and negative symptoms:* Positive and negative symptoms had stable, deteriorating, relapsing and ameliorating courses over time in patients, siblings and controls [19,20]. 

*Cognitive functions*: Cognitive subtypes had stable course during the six-year period in patients, siblings, controls and combined samples [21,22]. Additionally, cluster analyses showed “normal”, “mixed” and “impaired” cognitive profiles in siblings [23].

*Psychosocial functions*: Premorbid adjustment had “normal, slow decrease”, “normal, rapid decrease”, “mild, rapid decrease”, “mild, slow decrease”, “moderate, slow decrease” and “severe, slow decrease” longitudinal courses over the six-year period [26]. Another study showed “normal”, “social intermediate”, “academic decline”, “overall decline”, “overall intermediate” and “overall impaired” premorbid adjustment profiles in patients [25]. Multidimensional social inclusion had “very low (social functioning)/very low (quality of life)”, “low/low”, “high/low”, “medium/high” and “high/high” profiles [24]. 

### 3.3. Predictors of Latent Subtypes

*Positive and negative symptoms*: In patients, baseline positive symptom severity predicted latent subtypes of positive symptoms, while baseline negative symptom severity and quality of life predicted latent subtypes of negative symptoms [19]. In siblings, psychotic-like experiences (i.e., depressive symptoms) and quality of life predicted positive schizotypy subtypes, whereas premorbid adjustment, psychotic-like experiences (i.e., positive symptoms) and quality of life predicted negative schizotypy subtypes [19]. In controls, quality of life predicted positive schizotypy subtypes, while premorbid adjustment and quality of life predicted negative schizotypy subtypes.

*Cognitive functions*: Premorbid IQ, age and premorbid adjustment predicted cognitive latent subtypes in patients [21]. Age and poor functional outcomes predicted cognitive subtypes in siblings [21,23]. Interestingly, the cognitive symptoms of siblings were predicted by the cognitive subtypes of patients [21]. Age, premorbid IQ, premorbid adjustment and positive schizotypy predicted cognitive subtypes in controls [21]. PRS_SCZ,_ age, gender, ethnicity, educational status, premorbid IQ and premorbid adjustment predicted cognitive subtypes in combined samples [22]. 

*Psychosocial functions:* PRS_SCZ_, positive, negative and depressive symptoms, premorbid social functioning, number of met needs and satisfaction predicted multidimensional social inclusion subtypes [24]. 

## 4. Discussion

Through deep clinical phenotyping of SSD using the Genetic Risk and Outcome of Psychosis (GROUP) cohort dataset, we identified three to four latent subtypes of positive and negative symptoms [19,20] and four to six latent cognitive subtypes [21,22,23] in patients, siblings and controls. In our previous systematic review of worldwide longitudinal and cross-sectional data-driven studies, we observed two to five latent subtypes of positive, negative and cognitive symptoms in patients and controls [30]. In line with previous studies [31,32,33], we also identified five latent subtypes of psychosocial function—multidimensional social inclusion and premorbid adjustment—in patients [24,25,26]. Our findings support the DSM-5 dimensional approach for characterizing disorders based on a collection of symptoms [3]. Taken together, these findings demonstrate the inherent heterogeneity of SSD. Data-driven approaches have been shown to have the ability to explore heterogeneity and biological drivers of SSD and other psychotic disorders in the last decade [5]. Furthermore, data-driven approaches could contribute to the precise identification of high-risk population groups, optimal selection of interventions targeting subgroups of patients with similar phenotypes and evaluation of patients’ prognosis [34]. Consequently, they could help foster precision psychiatry by tackling diagnostic and treatment selection challenges pertaining to heterogeneity.

The identified latent subtypes in the GROUP studies [19,20,21,22,23,24,25,26] exhibited stable, deteriorating, improving and relapsing clinical courses over time with mixed phenotypes. We observed similar patterns of phenotypes in studies (particularly in patients and controls) included in our previous systematic review [30]. This implies the relevance of considering dynamic phenotypical changes over time, and therapies need to be seen as a process of change [35,36,37]. Interestingly, cognitive subtypes showed a stable course during the six-year follow-up [21,22]. This finding supports that cognitive deficits continue to be a valuable endophenotype of interest in the search for specific genetic factors related to schizophrenia [38,39]. The different symptom subtypes and their developmental courses appearing in both subclinical and clinical phases may elucidate the disease mechanisms before the disease onset and may be helpful for predicting transition to psychosis in unaffected individuals, which provides ample opportunities for early recognition, delaying onset and the treatment of subtle psychotic symptoms [40].

Even though up to 58 factors have been suggested to predict subtypes, only limited factors predicted latent subtypes in our studies [30]. This is possibly because we used multivariable models, whereas most of the other studies used univariable models or only pair-wise comparison was carried out [30]. Interestingly, we observed an overlap of factors that predict positive, negative and cognitive symptoms and psychosocial function subtypes. This suggests that these phenotypes have shared pathophysiology and similar clinical courses. Similarly, recently emerging evidence [41,42,43] showed a temporal association between positive, negative and cognitive symptoms and identified common predictive factors [44]. 

PRS_SCZ_ showed better association with cognitive subtypes than positive and negative symptom subtypes [21]. This suggests that cognition may have higher genetic susceptibility than positive and negative symptoms. Cognitive subtypes can be a suitable endophenotype of SSD [45] given the stable longitudinal course and its association with PRS_SCZ_. Endophenotype has been suggested to have a simpler genetic architecture than clinical syndromes and it can provide better signal-to-noise ratios and greater statistical power in research, which can be detectable in smaller sample sizes [45]. Furthermore, the strong predictive power of PRS_SCZ_ in the combined sample demonstrates the relevance of a large sample size and partly demonstrates the possibility of translating the constructed PRS_SCZ_ in research settings into actual practices to disentangle clinically relevant patient subtypes [46,47]. Broadly, the use of PRS is promising for early detection, the initiation of prevention strategies and the prediction of treatment outcomes, although studies that investigated the role of PRS_SCZ_ to predict latent subtypes are limited [30] and challenges exist in translational psychiatry research.

Heterogeneity in the traditional classification of mental disorders, including schizophrenia spectrum disorders (SSD), is the main challenge in research and clinical practice [48,49]. As such, key scientific values embodied in this article are highlighted as follows. Firstly, we combined data-driven, polygenic scoring and subphenotyping approaches to evaluate the profiles and longitudinal courses of SSD phenotypes by incorporating not only clinical but also psychosocial outcomes in patients and siblings. This is a unique aspect that has not been extensively explored in previous clinical studies. To best of our knowledge, there is no cohort study that comprehensively investigated positive and negative symptoms, cognitive impairments and social outcomes and compared findings across patients, siblings and controls. Most previous studies focused on positive and negative symptoms in patients, followed by cognitive impairment [30]. Studies that consistently examined the role of polygenic risk score to predict latent subtypes in positive and negative symptoms are very limited, let alone in social outcomes. Secondly, we depicted the presence of high heterogeneity and tried to show the full picture of SSD phenotypes, which is not addressed in the previous literature. Thirdly, we conducted comprehensive data analysis using various complementary statistical modelling techniques to obtain insight into the prominent predictors of latent subtypes. The combination of using data-driven approaches and the inclusion of PRS and exposomes enhances our understanding of SSD and its trajectory over time. Lastly, the novel findings presented in this article—particularly in the sibling population and psychosocial functioning phenotypes—can substantially contribute to the existing body of scientific knowledge and open new avenues for future investigations and potential clinical applications. 

The findings presented in this article also have clinical relevances and will contribute to a broader understanding of the clinical heterogeneity and disease course in SSD in two ways. First, data-driven methods can identify subgroups of SSD patients based on their phenotypes, which may ultimately help refine psychiatric nosology [49]. Additionally, this helps to reduce the heterogeneity for a more targeted assessment of clinical and biological predictors of these phenotypes in SSD. The large number of subtypes observed in studies also alerts clinicians to not use generic approaches to treat patients and consider siblings or family members during treatment. Second, the identification of specific biomarkers and clinical factors that are associated with specific latent subtypes will allow for the tailoring of treatments to alleviate positive and negative symptoms and improve cognitive performance and psychosocial functioning in SSD. We identified the predictors that can be used as a marker for clinicians for precisely identifying high-risk groups or patients with similar phenotypes. This could minimize the misdiagnosis that commonly seen in schizophrenia [13]. Overall, these findings may have the potential to impact patient care and early preventive and treatment strategies, leading to improved patient outcomes in the long term.

## 5. Strengths and Limitations

This article presented evidence on in-depth phenotyping of SSD targeting for multiple phenotypes in patients, siblings and controls. We identified latent subtypes using cross-sectional and longitudinal data. The longitudinal data also helped us to examine the long-term clinical course of phenotypes [13]. We also investigated various factors that predict latent subtypes and could be helpful for understanding disease progression. On the other hand, we presented findings based on a single cohort sample, which may limit the generalizability of our findings. However, our findings are comparable to results from a review of previous data-driven studies [30]. 

## 6. Future Research Directions

Data complexity, ambiguous subtyping results and nomenclature, and the complex relationship between phenotypes are still bottlenecks to fully harnessing the advantages of data-driven methods [12,48,50]. In the attempts to carry out in-depth phenotyping of SSD and the realization of precision psychiatric care, future collaborative research is warranted.

First, the identified latent subtypes and their clinical courses need to be validated in independent cohorts to ensure the generalizability of results. Additionally, more longitudinal studies are needed to evaluate the long-term course of cognitive and functional outcomes in patients and siblings. 

Second, subphenotyping targeting specific cognitive functions (e.g., memory, attention), positive symptoms (e.g., hallucination, delusion, disorganized thought), negative symptoms (e.g., asociality, avolition, affective flattening) and psychosocial functions (e.g., social engagement, housing, employment) in patients, siblings and controls using data-driven approaches may provide additional insights for deeper characterization of SSD. Endophenotype studies should also continue from interdisciplinary domains to address knowledge gaps and to deliver clinically applicable insights [45]. Additionally, more studies on siblings of patients with SSD are needed, as they are frequently affected due to shared genetic and non-genetic risk factors. For example, only 3 out of 53 worldwide data-driven studies were conducted on siblings [30]. Further data-driven studies on psychosocial function in patients and siblings are warranted.

Third, the use of growth mixture models or a dynamic mixture of expert models [51,52] may provide robust results, as these models allow the incorporation of both time-varying predictors and baseline predictors in the clustering of phenotypes and capture the dynamics better over time.

Fourth, as efforts have been devoted to unravelling the pathophysiology of SSD, it can also be beneficial to further examine the interrelatedness and distinctiveness of positive and negative symptoms, cognitive impairments and psychosocial functions [42] using causal discovery models [53,54]. 

Fifth, investigating the association of PRS_SCZ_ with phenotypes, endophenotypes, biological substrates and metabolites in large multinational consortia could help the identification of symptoms that have a stronger genetic liability and understanding of underlying causal models specific to clinical subgroups or symptom dimensions [55,56]. Performing GWAS of the trajectory of SSD phenotypes [57] and using the PRS of these phenotypes could be better than PRS_SCZ_ to deeply characterize latent subtypes [58,59]. Furthermore, calculating a polyomic risk score by integrating phenomics, genomics, epigenomics, proteomics, transcriptomics and pharmacogenomics may increase the performance of risk prediction, patients’ stratification for interventions and the efficacy of treatments [60,61].

Sixth, future studies may focus on enriching data-driven models based on clearly defined research questions and developing supplementary techniques for interpreting results and successfully translating findings into clinical practice [12,48]. It is also important to develop a pipeline composed of data collection, pre-processing and data modelling to identify subtypes with different profiles or disease progressions and risks of disease complications [18].

Seventh, future studies can be benefited by including inflammatory biomarkers, cardiometabolic data and lifestyle factors in patients and siblings, which could be critical for understanding the pathophysiology of somatic comorbidities that occur before or concomitantly with SSD [41]. In our previous study, we observed increased HbA1c levels in patients with SSD [62]. Using insulin sensitivity measures (euglycemic clamp), energy measurements and body composition could be also helpful for a more in-depth understanding of cardiometabolic co-morbidities that occur before and with SSD. Furthermore, extensive investigation of the role of antipsychotics and adherence to treatment could be helpful for precisely determining the clinical course of latent subtypes.

Eighth, applying Bayesian statistical models could provide strong evidence when the sample size is small, usability of omics data is limited, and follow-up duration is short compared to conventional statistical models. Bayesian inference is known to have an advantage given that the results are not dependent on the assumption of a large sample size, and it creates an opportunity to incorporate expert knowledge into the data-driven approach [63]. Therefore, the investigation of latent subtypes and their predictors using Bayesian latent class models can be an additional future research area [64,65].

Finally, recent advances in the theory and applicability of machine learning methods constitute an important area of future research due to the complexity of the data structure. Thus, hybrid methods that combine both supervised and unsupervised methods can identify subpopulations, contribute to comparing identified subpopulations and may provide superior results for the robust selection of predictors, which may ultimately help refine psychiatric nosology [49,66]. 

## 7. Conclusions

SSD phenotypes are highly heterogenous and show mixed profiles and longitudinal course among patients, siblings and controls. Three to four latent subtypes of positive and negative symptoms and four to six latent cognitive subtypes were identified in patients, siblings and controls. Five latent subtypes of psychosocial function—multidimensional social inclusion and premorbid adjustment—were identified in patients. We depicted that the identified subtypes had mixed profiles and stable, deteriorating, relapsing and ameliorating courses during the six-year follow-up period. Baseline positive and negative symptoms, premorbid adjustment, psychotic-like experiences, health-related quality of life and PRS_SCZ_ were found to be the strong predictors of the identified subtypes.

In the current era with limited objective criteria to diagnose SSD, the application of data-driven approaches could be relevant for clinicians and researchers for in-depth phenotyping of SSD by compiling all symptoms’ domains across patients and healthy individuals. The identification of latent subtypes and their clinical courses may help to design randomized control trials and implement targeted evidence-based interventions with maximized treatment efficacy and minimized costs and side effects. 

## Figures and Tables

**Figure 1 jpm-13-00954-f001:**
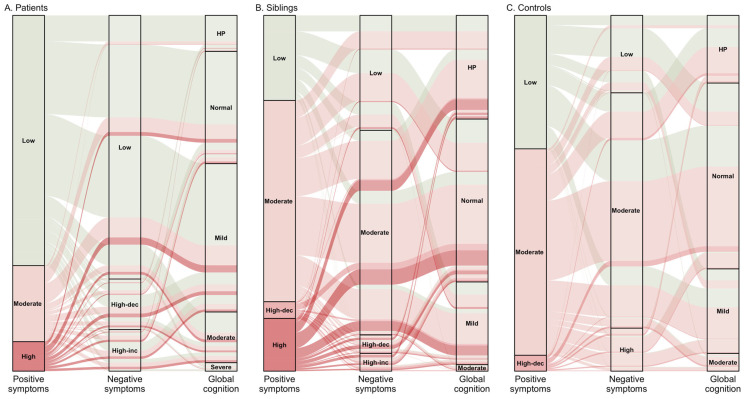
Spectrum of positive, negative and cognitive symptom subtypes in patients (**A**), siblings (**B**) and controls (**C**). High-dec: high-decreased; High-inc: high-increased; HP: high-performing. Cognitive subtypes coded from best to worse as high-performing, normal, mildly impaired, moderately impaired, severely impaired and very severely impaired.

**Figure 2 jpm-13-00954-f002:**
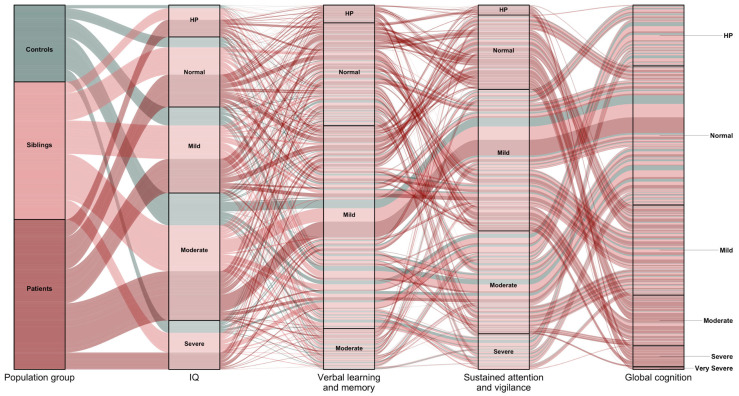
Spectrum of cognitive symptom subtypes in whole combined sample. HP: high-performing. Cognitive subtypes coded from best to worse as high-performing, normal, mildly impaired, moderately impaired, severely impaired and very severely impaired.

## Data Availability

Not applicable.

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
