# Peer review of "Deep Clinical Phenotyping of Schizophrenia Spectrum Disorders Using Data-Driven Methods: Marching towards Precision Psychiatry"

_jpm, 2023, doi:10.3390/jpm13060954_

Round 1
Reviewer 1 Report (Previous Reviewer 3)
jpm-2272033 : Habtewold et al: Marching towards precision psychiatry
The authors compared the results of their review of 53 Schizophrenia Spectrum Disorder (SSD) studies with the results of their longitudinal SSD study of 1,119 patients, 1,059 first-degree relatives, and 586 healthy control subjects, using several different instruments assessing positive, negative, and cognitive symptoms. They identified 2-6 latent subtypes that were comparable across studies and could be used to model the clinical course in the longitudinal study. They mentioned that the resulting trajectories were predictable by genetic, sociodemographic, social functioning, and disorder-related factors. And, the authors concluded that (1) cognitive deficits are a valuable endophenotype for genetic studies of schizophrenia; and (2) the findings are novel and of clinical interest for precisely identifying patients with good/poor prognosis.
Comments
Although the topic is of greater clinical interest, the manuscript remains very vague on almost all points. In particular, the reader is left in the dark as to how this manuscript differs from the author's earlier publications, or from the one that is currently under review.
The least the potential reader expects is a detailed description of the data material of the two studies, as well as their results. The manuscript gives practically no information on this matter.
In the methods section, the authors mention “linear regression”, “group-based trajectory modelling”, and “mixed-regression models” but do not go into the results in further detail as to statistical significance, or explained variance, amongst others.
Similar is the case with the hba1c data.
In its present form, this manuscript cannot be recommended for publication. It must be clearly worked out (1) in what way the manuscript differs from the earlier publications; (2) which data were evaluated in which form with which methods; and (3) how much of the observed variance was explained by the models used.
As to clinical relevance: the authors should demonstrate how their results may be translated into clinical practice.
The authors should substantiate why their proposed endophenotype for schizophrenia is more successful than those previously analyzed in the literature [e.g., Schizophr Res. 2016 ; 170(1): 30-40].
Author Response
Reviewer #1
The authors compared the results of their review of 53 Schizophrenia Spectrum Disorder (SSD) studies with the results of their longitudinal SSD study of 1,119 patients, 1,059 first-degree relatives, and 586 healthy control subjects, using several different instruments assessing positive, negative, and cognitive symptoms. They identified 2-6 latent subtypes that were comparable across studies and could be used to model the clinical course in the longitudinal study. They mentioned that the resulting trajectories were predictable by genetic, sociodemographic, social functioning, and disorder-related factors. And, the authors concluded that (1) cognitive deficits are a valuable endophenotype for genetic studies of schizophrenia; and (2) the findings are novel and of clinical interest for precisely identifying patients with good/poor prognosis.
First of all, we are very much grateful for your time and commitment to critically review our manuscript. The comments are helpful to improve the manuscript. Here, we presented our point-by-point reply to each comment or question. We hope we sufficiently addressed the comments/questions, and the revised manuscript is well-readable.
Comments
Although the topic is of greater clinical interest, the manuscript remains very vague on almost all points. In particular, the reader is left in the dark as to how this manuscript differs from the author's earlier publications, or from the one that is currently under review. The least the potential reader expects is a detailed description of the data material of the two studies, as well as their results. The manuscript gives practically no information on this matter.
Reply: This is an important suggestion. It is also pointed out by other reviewers. In this manuscript, our objective is to provide an overview of studies conducted by our research group on deep clinical phenotyping of schizophrenia and to share our insights to extend this line of research in the future. We believe the confusion partially due to Study 1 (review article) [1] and Study 5 (HbA1c study) [2]. Briefly, we excluded these two studies as their objectives were not in line with the other three studies (Studies 2 to 4) [3-5]. In the revised manuscript, we added an additional 5 studies [6-10], which had similar objectives with Studies 2 to 4. Study 1 [1] and Study 5 (HbA1c study) [2] used in the discussion section like any other cited references. Therefore, the revised manuscript is based on findings from 8 studies [3-10] on deep clinical phenotyping of schizophrenia targeting at positive and negative symptoms, cognitive impairment, and psychosocial functioning (page 2 of 16, lines 101 - 107). We also modified the results section and we explicitly and precisely present our main findings (Section 3, lines 163 - 230). To minimize unnecessary confusion, we cited the preprint [3] of the manuscript ‘under review’. We believe it is now clear that the current manuscript is different from the previous published systematic review (Study 1) and the manuscript ‘under review’.
In the methods section, the authors mention “linear regression”, “group-based trajectory modelling”, and “mixed-regression models” but do not go into the results in further detail as to statistical significance, or explained variance, amongst others. Similar is the case with the hba1c data.
Reply: In the revised version, we clearly presented the results of “group-based trajectory modelling” in sections 3.1. (Latent subtypes, lines 165 - 187) and 3.2. (Longitudinal course and profiles of subtypes, lines 198 - 209). The results of mixed-effects regression models (all significant predictors) were presented in section 3.3. (Predictors of latent subtypes, lines 211 - 229). We can not provide the detailed estimates of models as the objective of this manuscript is different and it is too much for this article. We kindly ask readers to read the original studies when they need extra information. We believe the result section is now precise, clear and easy to follow.
In its present form, this manuscript cannot be recommended for publication. It must be clearly worked out (1) in what way the manuscript differs from the earlier publications; (2) which data were evaluated in which form with which methods; and (3) how much of the observed variance was explained by the models used.
Reply: All the questions are relevant. As described before, our main objective in this manuscript is to briefly report the sequential findings of our three interlinked PhD studies over the past 10 years period (page 2 of 16, lines 101 - 107). We believe it sums up the evidence and make shortly available for clinicians and researchers. In the revised version, we reshuffled the methods (Section 2, lines 109-162) and results section (Section 3, lines 163 - 229), and we hope the difference is now clear. It is difficult to include detailed results due to possible duplication and other reasons mentioned above.
As to clinical relevance: the authors should demonstrate how their results may be translated into clinical practice.
Reply: In fact, it may be difficult for clinician to perform cluster analyses during health care provision. However, in our studies, we characterized the behavior of subgroups peoples (i.e., latent subtypes) and identified the predictors/markers. Therefore, clinicians can use these predictors/markers for precisely identifying high-risk population groups and patients with good/poor disease prognosis. Deep clinical phenotyping (clustering symptoms) could provide support for early detection of deteriorating patients, and to plan preventive interventions for different phenotypic groups, which ultimately results in determination of individualized and customized treatment and enhanced treatment outcome. We modified the clinical relevance as you may see in page 2 of 16, lines 284 - 293.
The authors should substantiate why their proposed endophenotype for schizophrenia is more successful than those previously analyzed in the literature [e.g., Schizophr Res. 2016 ; 170(1): 30-40].
Reply: We did not claim that our proposed endophenotype for schizophrenia is more successful than those previously analyzed in the literature. Our finding supports that cognitive deficits continue to be a valuable endophenotype of interest in the search for specific genetic factors related to schizophrenia and an indicator for clinical course of schizophrenia (page 2 of 16, 255 - 256).
References
- Habtewold TD, Rodijk LH, Liemburg EJ, Sidorenkov G, Boezen HM, Bruggeman R, et al. A systematic review and narrative synthesis of data-driven studies in schizophrenia symptoms and cognitive deficits. Translational Psychiatry. 2020;10(1). doi: 10.1038/s41398-020-00919-x.
- Habtewold TD, Islam MA, Liemburg EJ, Bruggeman R, Alizadeh BZ, Bartels-Velthuis AAA, et al. Polygenic risk score for schizophrenia was not associated with glycemic level (HbA1c) in patients with non-affective psychosis: Genetic Risk and Outcome of Psychosis (GROUP) cohort study. Journal of Psychosomatic Research. 2020;132:109968. doi: 10.1016/j.jpsychores.2020.109968.
- Habtewold TD, Tiles-Sar N, Liemburg EJ, Sandhu AK, Islam MA, Boezen HM, et al. Six-year trajectories and associated factors of positive and negative symptoms in schizophrenia patients, siblings, and controls: The GROUP study. 2023.
- Habtewold TD, Liemburg EJ, Islam MA, De Zwarte SMC, Boezen HM, Bruggeman R, et al. Association of schizophrenia polygenic risk score with data-driven cognitive subtypes: A six-year longitudinal study in patients, siblings and controls. Schizophrenia Research. 2020;223:135-47. doi: 10.1016/j.schres.2020.05.020.
- Islam MA, Habtewold TD, Van Es FD, Quee PJ, Van Den Heuvel ER, Alizadeh BZ, et al. Long-term cognitive trajectories and heterogeneity in patients with schizophrenia and their unaffected siblings. Acta Psychiatrica Scandinavica. 2018;138(6):591-604. doi: 10.1111/acps.12961.
- Hao J, Tiles-Sar N, Liemburg EJ, Habtewold TD, Bruggeman R, van der Meer L, et al. Multidimensional social inclusion and its prediction in schizophrenia spectrum disorder. 2023.
- Quee PJ, Alizadeh BZ, Aleman A, van den Heuvel ER. Cognitive subtypes in non-affected siblings of schizophrenia patients: characteristics and profile congruency with affected family members. Psychol Med. 2014;44(2):395-405. Epub 20130509. doi: 10.1017/s0033291713000809. PubMed PMID: 23659373.
- Quee PJ, Meijer JH, Islam MA, Aleman A, Alizadeh BZ, Meijer CJ, et al. Premorbid adjustment profiles in psychosis and the role of familial factors. Journal of Abnormal Psychology. 2014;123:578-87. doi: 10.1037/a0037189.
- Stiekema APM, Islam MA, Liemburg EJ, Castelein S, van den Heuvel ER, van Weeghel J, et al. Long-term course of negative symptom subdomains and relationship with outcome in patients with a psychotic disorder. Schizophr Res. 2018;193:173-81. Epub 20170622. doi: 10.1016/j.schres.2017.06.024. PubMed PMID: 28648915.
- Tiles-Sar N, Habtewold TD, Liemburg EJ, van der Meer L, Bruggeman R, Alizadeh BZ. Understanding Lifelong Factors and Prediction Models of Social Functioning After Psychosis Onset Using the Large-Scale GROUP Cohort Study. Schizophr Bull. 2023. Epub 20230427. doi: 10.1093/schbul/sbad046. PubMed PMID: 37104875.

Reviewer 2 Report (New Reviewer)
This study investigated the subtypes for schizophrenia. This research topic has both clinical interest and significance. The authors' presentation of this study and the overall study design, however, are quite muddled. Please see my main remarks below.
1. The METHOD part needs to be strengthened. Please keep in mind that the METHOD section needs to be brief, clear, and comprehensive enough to let other researchers assess the level of quality and validity of the study. The current approach, however, did not offer enough details for the data analysis. For instance, according to the current description, I am unable to determine whether the authors used the findings from earlier studies or independently determined the subtypes. In any case, I think the authors ought to state this explicitly and at the very least describe the statistical techniques they employed to identify latent subtypes. Another example is the polygenic risk score of SCZ; in this case, the authors should at the very least provide the information required to calculate the polygenic risk score, such as the underlying GWAS.
2. The authors included the findings of a different study twice in the RESULT section and simply stated that it was "under review" without providing any further details. First off, I'm not sure if this is the proper way to cite a manuscript in this journal. I think it may be preferable to cite a preprint for an unpublished paper. Alternatively, in my opinion, the authors should give the reviewer the pertinent details of this unpublished paper.
3. To me, it appears that the authors simply listed earlier findings in section 3.3 without adding anything new.
4. It is also quite unclear why the authors looked into glycaemic dysregulation in schizophrenia and the risk factors for it. The INTRODUCTION offers no context or explanation.
5. In the meanwhile, the authors should list pertinent references when they describe the background of this investigation. For example, the authors claimed that cognitive deficits barely respond to antipsychotic treatment without citing supporting references.
Author Response
Reviewer #2
This study investigated the subtypes for schizophrenia. This research topic has both clinical interest and significance. The authors' presentation of this study and the overall study design, however, are quite muddled. Please see my main remarks below.
First of all, we are very much grateful for your time and commitment to critically review our manuscript. The comments are helpful to improve the manuscript. Here, we presented our point-by-point reply to each comment or question. We hope we sufficiently addressed the comments/questions, and the revised manuscript is well-readable.
- The METHOD part needs to be strengthened. Please keep in mind that the METHOD section needs to be brief, clear, and comprehensive enough to let other researchers assess the level of quality and validity of the study. The current approach, however, did not offer enough details for the data analysis. For instance, according to the current description, I am unable to determine whether the authors used the findings from earlier studies or independently determined the subtypes. In any case, I think the authors ought to state this explicitly and at the very least describe the statistical techniques they employed to identify latent subtypes. Another example is the polygenic risk score of SCZ; in this case, the authors should at the very least provide the information required to calculate the polygenic risk score, such as the underlying GWAS.
Reply: This is an important suggestion. It is also pointed out by the other reviewers. In this manuscript, our objective is to provide an overview of studies conducted by our research group on deep clinical phenotyping of schizophrenia and to share our insights to extend this line of research in the future. Therefore, we refrained from providing detailed information in the methods section, which have been published and discussed in the original studies. We kindly ask readers to refer the original studies when they need extra information. In the revised version, we briefly described statistical techniques employed to identify latent subtypes (page 1 of 16, lines 150 - 162) and polygenic risk score (page 1 of 16, lines 146 - 149) followed by citing relevant references. We believe that the confusion partially due to Study 1 (review article) [1]. We used the findings from earlier studies in Study 1 (review article) [1], however, we independently identified the latent subtypes in Studies 2 to 4 [2-4] using our cohort data. In the revised version, we excluded Study 1 [1] and Study 5 [5] because their objectives are not in line with the other three studies (Studies 2 to 4) [2-4]. We added an additional 5 studies [6-10], which had similar objectives with Studies 2 to 4 and used a similar cohort data. Therefore, the revised manuscript is based on findings from 8 studies [2-4, 6-10] on deep clinical phenotyping of schizophrenia targeting at positive and negative symptoms, cognitive impairment, and psychosocial functioning (page 3 of 16, lines 101 - 107).
- The authors included the findings of a different study twice in the RESULT section and simply stated that it was "under review" without providing any further details. First off, I'm not sure if this is the proper way to cite a manuscript in this journal. I think it may be preferable to cite a preprint for an unpublished paper. Alternatively, in my opinion, the authors should give the reviewer the pertinent details of this unpublished paper.
Reply: We agree. For the study “under review’, we cited the preprint [2] to minimize unnecessary confusion.
- To me, it appears that the authors simply listed earlier findings in section 3.3 without adding anything new.
Reply: We intentionally focused on presenting and linking the main findings of our several original studies done only by our research group on deep clinical phenotyping of schizophrenia and sharing our insights to extend this line of research. We clarified this in the introduction section as you may see in page 3 of 16, lines 101 – 107. This article is not a formal systematic review as you may saw from the method section.
- It is also quite unclear why the authors looked into glycaemic dysregulation in schizophrenia and the risk factors for it. The INTRODUCTION offers no context or explanation.
Reply: We agree. As we described before, we excluded Study 5 [5] (glycaemic dysregulation, HbA1c study) given that the objective was not in line with the other studies objectives. Therefore, the introduction about the context is not anymore relevant. Study 5 [5] used only in the discussion section for illustration like any other references.
- In the meanwhile, the authors should list pertinent references when they describe the background of this investigation. For example, the authors claimed that cognitive deficits barely respond to antipsychotic treatment without citing supporting references.
Reply: We thoroughly revised the introduction section and cited relevant references (Section 1, lines 39 - 107). This also noted by the other reviewers.
References
- Habtewold TD, Rodijk LH, Liemburg EJ, Sidorenkov G, Boezen HM, Bruggeman R, et al. A systematic review and narrative synthesis of data-driven studies in schizophrenia symptoms and cognitive deficits. Translational Psychiatry. 2020;10(1). doi: 10.1038/s41398-020-00919-x.
- Habtewold TD, Tiles-Sar N, Liemburg EJ, Sandhu AK, Islam MA, Boezen HM, et al. Six-year trajectories and associated factors of positive and negative symptoms in schizophrenia patients, siblings, and controls: The GROUP study. 2023.
- Habtewold TD, Liemburg EJ, Islam MA, De Zwarte SMC, Boezen HM, Bruggeman R, et al. Association of schizophrenia polygenic risk score with data-driven cognitive subtypes: A six-year longitudinal study in patients, siblings and controls. Schizophrenia Research. 2020;223:135-47. doi: 10.1016/j.schres.2020.05.020.
- Islam MA, Habtewold TD, Van Es FD, Quee PJ, Van Den Heuvel ER, Alizadeh BZ, et al. Long-term cognitive trajectories and heterogeneity in patients with schizophrenia and their unaffected siblings. Acta Psychiatrica Scandinavica. 2018;138(6):591-604. doi: 10.1111/acps.12961.
- Habtewold TD, Islam MA, Liemburg EJ, Bruggeman R, Alizadeh BZ, Bartels-Velthuis AAA, et al. Polygenic risk score for schizophrenia was not associated with glycemic level (HbA1c) in patients with non-affective psychosis: Genetic Risk and Outcome of Psychosis (GROUP) cohort study. Journal of Psychosomatic Research. 2020;132:109968. doi: 10.1016/j.jpsychores.2020.109968.
- Hao J, Tiles-Sar N, Liemburg EJ, Habtewold TD, Bruggeman R, van der Meer L, et al. Multidimensional social inclusion and its prediction in schizophrenia spectrum disorder. 2023.
- Quee PJ, Alizadeh BZ, Aleman A, van den Heuvel ER. Cognitive subtypes in non-affected siblings of schizophrenia patients: characteristics and profile congruency with affected family members. Psychol Med. 2014;44(2):395-405. Epub 20130509. doi: 10.1017/s0033291713000809. PubMed PMID: 23659373.
- Quee PJ, Meijer JH, Islam MA, Aleman A, Alizadeh BZ, Meijer CJ, et al. Premorbid adjustment profiles in psychosis and the role of familial factors. Journal of Abnormal Psychology. 2014;123:578-87. doi: 10.1037/a0037189.
- Stiekema APM, Islam MA, Liemburg EJ, Castelein S, van den Heuvel ER, van Weeghel J, et al. Long-term course of negative symptom subdomains and relationship with outcome in patients with a psychotic disorder. Schizophr Res. 2018;193:173-81. Epub 20170622. doi: 10.1016/j.schres.2017.06.024. PubMed PMID: 28648915.
- Tiles-Sar N, Habtewold TD, Liemburg EJ, van der Meer L, Bruggeman R, Alizadeh BZ. Understanding Lifelong Factors and Prediction Models of Social Functioning After Psychosis Onset Using the Large-Scale GROUP Cohort Study. Schizophr Bull. 2023. Epub 20230427. doi: 10.1093/schbul/sbad046. PubMed PMID: 37104875.

Reviewer 3 Report (New Reviewer)
This is an interesting study conducted in different sample groups targeting hallmark symptoms of schizophrenia spectrum disorders. The authors showed two to six latent subtypes of positive, negative and cognitive symptoms in patients, siblings and controls through extensive analyses of published studies and GROUP cohort data analyses. Their findings seem to be novel and of clinical interest for precisely identifying patients with good/poor disease prognosis and high-risk population groups, which could advance the knowledge in the field.
Author Response
Reviewer #3
This is an interesting study conducted in different sample groups targeting hallmark symptoms of schizophrenia spectrum disorders. The authors showed two to six latent subtypes of positive, negative and cognitive symptoms in patients, siblings and controls through extensive analyses of published studies and GROUP cohort data analyses. Their findings seem to be novel and of clinical interest for precisely identifying patients with good/poor disease prognosis and high-risk population groups, which could advance the knowledge in the field.
Reply: We are very much grateful for your time and commitment to critically review our manuscript. We are also thankful for the kind words and complements.
Reviewer 4 Report (New Reviewer)
Needs editing throughout to improve clarity and syntax. Authors need to be more restrained in their conclusions and specific in their suggestions.
Intro
57: wouldn’t describe positive symptoms, negative symptoms, and cognitive deficits as “phenotypes”…consider “features”?
58: says “probands and general population”—this seems to be a typo or a word is missing. Do you mean to say “compared to”?
61-62: suggest rewording, e.g., “…by schizophrenia and are associated with 15-30 years of life lost in persons with schizophrenia compared to…”
63: delete “solely”. Clinical experience/wisdom and family history also inform dx process.
71: “diagnostic” should be singular
75-76: should this say that comorbidity is particularly high/prevalence in elderly adults with scz? This is true across natural aging, not just SMI.
76: “minimizes” seems to be the wrong word here. Do you mean “applicability”? what does the field of geriatric psychiatry say?
89: replace till with “until” (till/til is slang).
92-93: sociodemographic diversity in epidemiologic studies is NOT a limitation of them. Please clarify this sentence.
101: should this read latent subtypes/trajectories of PEOPLE according to their symptoms? There are different types of latent modeling; some are variable-centered and some are person-centered. Be clear about which you conducted.
Methods:
135-136: did you do this at baseline or longitudinally? Use of GBTM (lines 165-166) would suggest that you identified subgroups of people based on their patterns of symptoms over time. If different sets of latent modeling were conducted, this needs to be made clearer. Readers shouldn’t have to look up the primary papers to clarify the methods you used.
Results
177: cool figure. Please include the method you used for creating it (e.g., software/package). These findings seem to show an incredible amount of heterogeneity, with clustering in the expected directions, but I am unsure how this solves the problem with dx.
196-197: did you look at whether those with any mood/anx comorbidity or related diagnostic instability (e.g., someone prev dxed with sczaffective, bipolar, depression) were more likely to be in a fluctuating course?
207-210: this is very interesting. Would it suggest a potential sequence of depression (precursor) followed by positive symptoms, at which point risk for negative symptoms increases?
222: just say the number or % instead of “moderate level” which can mean anything to anyone
Discussion
238-250: what of this is backed up by data (cite it) vs what is opinion? These are the hopes of personalized medicine and they have NOT materialized yet for psychiatry.
253: “which can” seems to be missing a word
277-278: was genetic data available for all participants? If not, immigrative selection bias may limit the strength of this conclusion.
281: irregular use of the word analogous, please correct
282: this is an overreach. You didn’t study treatment response (although that would be a natural and interesting follow-up).
283-287: Or, maybe PRS predicts cognitive trajectories better because the people in the original studies (did you use PGC data to identify disease-associated SNPs?) may have had more cognitive symptoms (it could have driven them to join the studies), which would in turn mean the SNPs and resultant risk scores are going to be better for cog symptoms.
329-330: needs minor revision, maybe should say something like “…directions, with stakeholders actively and collaboratively working toward the realization…”
331 onward: I would suggest beginning your future directions with the areas you identified as limitations of your own work (lines 316-322). So, move “second” (replication and external validation) and “sixth” (examine alternative latent indicators to assess the stability of your findings) up to the top, before beginning to go in other directions.
339: why didn’t you use GMM in the first place?
346: reword this sentence.
365: is “effective” the word you want here?
373: I realized ML is very trendy right now, but something important has been missed. Latent classification is itself a form of unsupervised ML. if you want to discuss ML here, be specific rather than general/superficial.
380: consider beginning with a summative sentence.
381: replace patients with persons with scz, or similar wording.
390: “decidedly relevalent” ...for what and to whom?
392: it would be premature to use any of this for designing RCTs! Again, remember the limitations of your study …the replication/validation that still needs to be done, as well as the need to examine stability of findings across alternative indicators.
394-396: That is not what you tested. You did not compare dimensions vs discrete. The use of continuous indicators is not analogous to supporting dimensionality.
Author Response
Reviewer #4
Needs editing throughout to improve clarity and syntax. Authors need to be more restrained in their conclusions and specific in their suggestions.
First of all, we are very much grateful for your time and commitment to critically review our manuscript. Here, we presented our point-by-point reply to each comment or question. We hope we sufficiently addressed the comments/questions and the revised manuscript is well-readable.
Introduction
57: wouldn’t describe positive symptoms, negative symptoms, and cognitive deficits as “phenotypes”…consider “features”?
Reply: We understand that “phenotypes”, “symptoms”, “features” are interchangeably used in literature. In clinical medicine, “phenotype” commonly used to describe diseases symptoms, and we want to keep it to minimize unnecessary confusion for clinicians.
58: says “probands and general population”—this seems to be a typo or a word is missing. Do you mean to say “compared to”?
Reply: We revised the sentence as follows, “It is noteworthy to indicate that these phenotypes are also diversely present in siblings due to shared genetic and non-genetic factors with their probands.” (page 2 of 16, lines 62 - 63)
61-62: suggest rewording, e.g., “…by schizophrenia and are associated with 15-30 years of life lost in persons with schizophrenia compared to…”
Reply: The purpose of this sentence was to show the context about T2D comorbidity in patients with schizophrenia. The whole sentence was removed as we removed study 5 (T2D comorbidity)[1] in the revised manuscript as its objective is different from the other included studies. Therefore, it is not anymore important.
63: delete “solely”. Clinical experience/wisdom and family history also inform dx process.
Reply: We deleted the word ‘solely’ as you may see in page 2 of 16, lines 64 - 67.
71: “diagnostic” should be singular
Reply: We replaced ‘diagnostics’ with ‘diagnostic’ as you may see in page 2 of 16, line 73.
75-76: should this say that comorbidity is particularly high/prevalence in elderly adults with scz? This is true across natural aging, not just SMI.
Reply: We deleted the whole sentence to minimize unnecessary confusion.
76: “minimizes” seems to be the wrong word here. Do you mean “applicability”? what does the field of geriatric psychiatry say?
Reply: We deleted the whole sentence to minimize unnecessary confusion.
89: replace till with “until” (till/til is slang).
Reply: We replaced ‘till’ with ‘until’ as you may see in page 2 of 16, line 89.
92-93: sociodemographic diversity in epidemiologic studies is NOT a limitation of them. Please clarify this sentence.
Reply: We deleted the phrase “the sociodemographic difference of study participants” as you may see page 2 of 16, line 92 - 95.
101: should this read latent subtypes/trajectories of PEOPLE according to their symptoms? There are different types of latent modeling; some are variable-centered and some are person-centered. Be clear about which you conducted.
Reply: We used patient-centered latent class modeling to identify latent subtypes/trajectories of people according to their symptoms level. We clarified this in the methods section as you may see in page 1 of 16, lines 151 - 153.
Methods:
135-136: did you do this at baseline or longitudinally? Use of GBTM (lines 165-166) would suggest that you identified subgroups of people based on their patterns of symptoms over time. If different sets of latent modeling were conducted, this needs to be made clearer. Readers shouldn’t have to look up the primary papers to clarify the methods you used.
Reply: In GROUP cohort, data was collected at baseline, and 3-year and 6-year follow-up. In 5 studies [2-6], we used GBTM to identify subgroups of people based on their patterns of symptoms over time (i.e., entire 6 years follow-up), whereas we applied K-means clustering in three studies [7-9] based on cross-sectional data available (baseline data, 3-year, or 6-year). We clarified this in the methods section as you may see in page 1 of 16, lines 120 - 149.
Results
177: cool figure. Please include the method you used for creating it (e.g., software/package). These findings seem to show an incredible amount of heterogeneity, with clustering in the expected directions, but I am unsure how this solves the problem with dx.
Reply: We created Figure 1 in R software using ggplot2 package. We described this in the method section as you may see in page 2 of 16, lines 157 and 162. The current diagnosis approach is mainly based on the set of DSM criteria and the decision is only yes – diseased or no – not diseased. However, in our studies, we showed the presence of large heterogeneity even within the patient group. One patient may have different severity of symptoms despite fulfilling the criteria for disease diagnosis. Therefore, our results could help to precisely identify individuals with similar level symptoms/diseases severity, which ultimately minimize mis- or over-diagnosis.
196-197: did you look at whether those with any mood/anx comorbidity or related diagnostic instability (e.g., someone prev dxed with sczaffective, bipolar, depression) were more likely to be in a fluctuating course?
Reply: This is important question. Depression symptoms during follow-up significantly predicted latent subtypes of positive and negative symptoms with fluctuating course [3]. Only limited number of patients had previous history of other mental illness and we did not test this.
207-210: this is very interesting. Would it suggest a potential sequence of depression (precursor) followed by positive symptoms, at which point risk for negative symptoms increases?
Reply: Thank you. This could be possible, but it needs further investigation.
222: just say the number or % instead of “moderate level” which can mean anything to anyone
Reply: We deleted the whole findings of study 5 [1] as described before.
Discussion
238-250: what of this is backed up by data (cite it) vs what is opinion? These are the hopes of personalized medicine and they have NOT materialized yet for psychiatry.
Reply: These are our views. We revised this section to loosen it up our hopes as you may see in page 1, lines 318 – 337.
253: “which can” seems to be missing a word
Reply: We deleted the phrase “during which can be readjusted” to minimize unnecessary confusion.
277-278: was genetic data available for all participants? If not, immigrative selection bias may limit the strength of this conclusion.
Reply: Yes, genetic data is available for all participants.
281: irregular use of the word analogous, please correct
Reply: We reshuffled the last part of the sentence as you may see in page 1 of 16, line 267 - 269.
282: this is an overreach. You didn’t study treatment response (although that would be a natural and interesting follow-up).
Reply: We moved this sentence to the ‘future perspective’ section as you may see in page 3 of 16, lines 354 - 356.
283-287: Or, maybe PRS predicts cognitive trajectories better because the people in the original studies (did you use PGC data to identify disease-associated SNPs?) may have had more cognitive symptoms (it could have driven them to join the studies), which would in turn mean the SNPs and resultant risk scores are going to be better for cog symptoms.
Reply: This is an interesting suggestion. It could be one of the reasons. We used PGC-GWAS summary statistics to calculate the polygenic risk score of schizophrenia. However, we are unable to confirm the severity of cognitive symptoms in PGC samples.
329-330: needs minor revision, maybe should say something like “…directions, with stakeholders actively and collaboratively working toward the realization…”
Reply: We revised based on the suggestion as you may see in page 2 of 16, line 307 - 309.
331 onward: I would suggest beginning your future directions with the areas you identified as limitations of your own work (lines 316-322). So, move “second” (replication and external validation) and “sixth” (examine alternative latent indicators to assess the stability of your findings) up to the top, before beginning to go in other directions.
Reply: We rearranged base on the suggestion as you may see in the ‘future perspectives’ section as you may see in page 2 of 16, lines 310 - 324.
339: why didn’t you use GMM in the first place?
Reply: We did not use GMM because of lack experts at the time of the study.
346: reword this sentence.
Reply: We modified the fourth recommendation as you may see in page 2 of 16, lines 329 – 332.
365: is “effective” the word you want here?
Reply: We deleted the word “effective” as you may see in page 3 of 16, lines 357 - 363.
373: I realized ML is very trendy right now, but something important has been missed. Latent classification is itself a form of unsupervised ML. if you want to discuss ML here, be specific rather than general/superficial.
Reply: Yes, indeed. We revised reshuffled the final recommendation on ML as you may see in page 3 of 16, lines 364 - 368.
380: consider beginning with a summative sentence.
Reply: We revised the first part of the conclusion section based on the suggestion as you may see in page 3 of 16, lines –370 - 378.
381: replace patients with persons with scz, or similar wording.
Reply: In the method section, we described that patients were diagnosed with schizophrenia-spectrum disorders, and we used only ‘patients’ throughout the manuscript to minimize redundancy as well as to save some space.
390: “decidedly relevalent” ...for what and to whom?
Reply: We clarified this as you may see in page 3 of 16, lines 379 - 384.
392: it would be premature to use any of this for designing RCTs! Again, remember the limitations of your study …the replication/validation that still needs to be done, as well as the need to examine stability of findings across alternative indicators.
Reply: We modified this sentence to loosen up our recommendation as you may see in page 4 of 16, lines 382 - 384.
394-396: That is not what you tested. You did not compare dimensions vs discrete. The use of continuous indicators is not analogous to supporting dimensionality.
Reply: It is true that we did not compare dimensional approach with categorical approach. However, we showed that deep phenotyping based on severity of clinical symptoms (i.e., “how much” question of DSM-5) helped us identify multiple subgroups of individuals in patients and health samples (i.e., siblings and controls. The dimensional approach allows a clinician to assess the severity of symptoms with numerical values on one or more scales or continuums, as a result profile/subtype is created instead of labelling and help to characterize baseline clinical status and subsequent changes in clinical status with treatment [10]. For example, in our studies lower positive and negative symptoms scores equate to lower impairment and higher scores equate to higher impairment. In general, as we described in the introduction section, counting number of symptoms (based on presence or absence, prone to subjective interpretations) is not providing sufficient information and symptoms severity need to be quantified and used to characterize individuals.
References
- Habtewold TD, Islam MA, Liemburg EJ, Bruggeman R, Alizadeh BZ, Bartels-Velthuis AAA, et al. Polygenic risk score for schizophrenia was not associated with glycemic level (HbA1c) in patients with non-affective psychosis: Genetic Risk and Outcome of Psychosis (GROUP) cohort study. Journal of Psychosomatic Research. 2020;132:109968. doi: 10.1016/j.jpsychores.2020.109968.
- Habtewold TD, Liemburg EJ, Islam MA, De Zwarte SMC, Boezen HM, Bruggeman R, et al. Association of schizophrenia polygenic risk score with data-driven cognitive subtypes: A six-year longitudinal study in patients, siblings and controls. Schizophrenia Research. 2020;223:135-47. doi: 10.1016/j.schres.2020.05.020.
- Habtewold TD, Tiles-Sar N, Liemburg EJ, Sandhu AK, Islam MA, Boezen HM, et al. Six-year trajectories and associated factors of positive and negative symptoms in schizophrenia patients, siblings, and controls: The GROUP study. 2023.
- Islam MA, Habtewold TD, Van Es FD, Quee PJ, Van Den Heuvel ER, Alizadeh BZ, et al. Long-term cognitive trajectories and heterogeneity in patients with schizophrenia and their unaffected siblings. Acta Psychiatrica Scandinavica. 2018;138(6):591-604. doi: 10.1111/acps.12961.
- Stiekema APM, Islam MA, Liemburg EJ, Castelein S, van den Heuvel ER, van Weeghel J, et al. Long-term course of negative symptom subdomains and relationship with outcome in patients with a psychotic disorder. Schizophr Res. 2018;193:173-81. Epub 20170622. doi: 10.1016/j.schres.2017.06.024. PubMed PMID: 28648915.
- Tiles-Sar N, Habtewold TD, Liemburg EJ, van der Meer L, Bruggeman R, Alizadeh BZ. Understanding Lifelong Factors and Prediction Models of Social Functioning After Psychosis Onset Using the Large-Scale GROUP Cohort Study. Schizophr Bull. 2023. Epub 20230427. doi: 10.1093/schbul/sbad046. PubMed PMID: 37104875.
- Hao J, Tiles-Sar N, Liemburg EJ, Habtewold TD, Bruggeman R, van der Meer L, et al. Multidimensional social inclusion and its prediction in schizophrenia spectrum disorder. 2023.
- Quee PJ, Alizadeh BZ, Aleman A, van den Heuvel ER. Cognitive subtypes in non-affected siblings of schizophrenia patients: characteristics and profile congruency with affected family members. Psychol Med. 2014;44(2):395-405. Epub 20130509. doi: 10.1017/s0033291713000809. PubMed PMID: 23659373.
- Quee PJ, Meijer JH, Islam MA, Aleman A, Alizadeh BZ, Meijer CJ, et al. Premorbid adjustment profiles in psychosis and the role of familial factors. Journal of Abnormal Psychology. 2014;123:578-87. doi: 10.1037/a0037189.
- Association AP. DSM-5's integrated approach to diagnosis and classifications. Arlington, VA: American Psychiatric Association. 2013.

Reviewer 5 Report (New Reviewer)
This perspective presents commentary on a set of articles looking to identify a set of variables that predict SSD and their subtypes. It is a good, focused review on the topic pertaining to studies (including a previous review) conducted by the authors. A few points to consider:
-in your future directions, are there other omic fields that should be investigated that could be used to create a "poly-omic risk score" that goes beyond the polygenic risk score methodology your studies explored? Do you believe epigenetics could be a key variable in reflecting both environmental and inter-individual risk stratification?
-Is it possible that key to understanding cardiometabolic co-morbidities that occur before and with SSD could be more in-depth phenotyping beyond simple glucose/A1C measurements such as insulin sensitivity measures (euglycemic clamp), energy measurements, body composition, etc
-How do medications (antipsychotics) play a role in your studies and in future work?
Author Response
Reviewer #5
This perspective presents commentary on a set of articles looking to identify a set of variables that predict SSD and their subtypes. It is a good, focused review on the topic pertaining to studies (including a previous review) conducted by the authors.
First of all, we are very much grateful for your time and commitment to critically review our manuscript. The comments are very helpful to improve the manuscript. Here, we presented our point-by-point reply to each comment or question. We hope we sufficiently addressed the comments/questions and the revised manuscript is well-readable.
A few points to consider:
-in your future directions, are there other omic fields that should be investigated that could be used to create a "poly-omic risk score" that goes beyond the polygenic risk score methodology your studies explored? Do you believe epigenetics could be a key variable in reflecting both environmental and inter-individual risk stratification?
Reply: Both are very important points for deep phenotyping of schizophrenia spectrum disorders. In our cohort, we have genomic data for all participants and metabolomics data only for patients. Epigenetics is a key variable as well for individual risk stratification. We included both points in our future direction section (page 3 of 16, lines 333 - 341).
-Is it possible that key to understanding cardiometabolic co-morbidities that occur before and with SSD could be more in-depth phenotyping beyond simple glucose/A1C measurements such as insulin sensitivity measures (euglycemic clamp), energy measurements, body composition, etc
Reply: This is important question. We used simple glucose/HbA1C measurements because that was the only data collected in our cohort to assess glucose dysregulation or diabetes. We included this suggestion in our ‘future perspectives’ section as you may see in page 3 of 16, lines 348 - 354
-How do medications (antipsychotics) play a role in your studies and in future work?
Reply: The role of antipsychotics was not extensively investigated in our studies because of lack of sufficient medication data. Study 1 (HbA1c study) showed that high metabolic risk antipsychotics were significantly associated with increased HbA1c levels. Currently, we are working on several projects investigating the role of antipsychotics on risk of cardiometabolic comorbidities, and whether they are improving cognitive and psychosocial functioning in patients with schizophrenia. We included this suggestion in our ‘future perspectives’ section as you may see in page 3 of 16, lines 354 - 356.
Reviewer 6 Report (New Reviewer)
Introduction
Different sentences of the introduction need more references (beginning of page 2).
The criticism of the DSM nosographic system is very explicit and based on very few bibliographic references. The authors do not consider the positive aspects of the DSM, which is the most widespread nosographic manual used in international scientific and clinical communities.
The following sentence is also very generic and is based on a single reference. “For instance, schizophrenia is particularly comorbid in elderly patients, which minimizes treatment guidelines, as often not ‘real patients’ but ‘ideal patients’ were focused on in randomized control trials and the guideline development (13)”.
The authors stated four objectives; however, the study hypotheses are lacking.
Methods
The entire paper is a review of 4 articles published by one or more of the authors of this manuscript. The authors did not state the methods of selection of the included articles, and it is unclear why other studies on the same topic are not considered. The authors reported the methods followed by the reviewed studies, one of which is a literature review.
Results
Results are based exclusively on the above four articles and on a study under review (In patients, baseline positive symptom severity was the predictor of latent subtypes of positive symptoms, while baseline negative symptom severity and quality of life were predictors of latent subtypes of negative symptoms [Under review])
Discussion
The discussion reports a mixture of evidence concerning the polygenic risk and some psychopathological and pathophysiological aspects (hba1c levels) related to the four included articles. However, it appears more appropriate for a book chapter than a scientific article.
The “Strengths and limitations” paragraph is not related to the manuscript but mainly reports the strengths and limitations of the four included studies.
Author Response
Reviewer #6
First of all, we are very much grateful for your time and commitment to critically review our manuscript. The comments are helpful to improve the manuscript. Here, we presented our point-by-point reply to each comment or question. We hope we sufficiently addressed the comments/questions, and the revised manuscript is well-readable.
Introduction
Different sentences of the introduction need more references (beginning of page 2).
Reply: We thoroughly checked the whole introduction and reference added when necessary (Section 1, lines 39 - 107).
The criticism of the DSM nosographic system is very explicit and based on very few bibliographic references. The authors do not consider the positive aspects of the DSM, which is the most widespread nosographic manual used in international scientific and clinical communities.
Reply: We totally agree. We included both the strengths and drawbacks of DSM nosography system as you may see in page 2 of 16, lines 64 - 73.
The following sentence is also very generic and is based on a single reference. “For instance, schizophrenia is particularly comorbid in elderly patients, which minimizes treatment guidelines, as often not ‘real patients’ but ‘ideal patients’ were focused on in randomized control trials and the guideline development (13)”.
Reply: We agree. We removed this sentence to minimize unnecessary confusion. It is less relevant as a background information.
The authors stated four objectives; however, the study hypotheses are lacking.
Reply: In this article, we focused on presenting main findings studies done by our research group and sharing our insights to extend this line of research. We are restrained ourselves to repeat several aspects and extensive details of the original studies on which this overarching review is based. Furthermore, we implemented hypothesis free data-driven methods and specific hypotheses on predictors were presented in the original studies.
Methods
The entire paper is a review of 4 articles published by one or more of the authors of this manuscript. The authors did not state the methods of selection of the included articles, and it is unclear why other studies on the same topic are not considered. The authors reported the methods followed by the reviewed studies, one of which is a literature review.
Reply: This important suggestion. This article is not a systematic review as you may saw from the method section. As we described before, our main objective in this manuscript is to briefly report the sequential findings of our three interlinked PhD studies over the past 10 years period. We believe it sums up the evidence and make shortly available for clinicians and researchers. We think the confusion partially comes from Study 1 (review article) [1]. Briefly, we excluded our literature review article (Study 1) [1] to minimize unnecessary confusion. In the revised version, we added an additional 5 studies [2-6]. Thus main findings from 8 studies [2-9] on deep clinical phenotyping of schizophrenia targeting at positive and negative symptoms, cognitive impairment, and psychosocial functioning were summarized. We clarified this in the introduction section (page 3 of 16, lines 101 - 107).
Results
Results are based exclusively on the above four articles and on a study under review (In patients, baseline positive symptom severity was the predictor of latent subtypes of positive symptoms, while baseline negative symptom severity and quality of life were predictors of latent subtypes of negative symptoms [Under review])
Reply: As described before, the revised manuscript is based on findings from a total of 8 studies, which had similar objectives (page 3 of 16, lines 101 - 107). To minimize unnecessary confusion, we cited the preprint [7] of the manuscript ‘under review’. This is also noted by the other reviewers.
Discussion
The discussion reports a mixture of evidence concerning the polygenic risk and some psychopathological and pathophysiological aspects (hba1c levels) related to the four included articles. However, it appears more appropriate for a book chapter than a scientific article.
Reply: In the revised manuscript, we removed the study on hba1c (Study 5) [10] and any texts related to comorbidity, except in the ‘Future research direction’ section. Additionally, we reshuffled the whole discussion texts. We would like again to note this manuscript is formed to present summary of the consistent finding overarching three PhD theses from our group over the past 10 years period.
The “Strengths and limitations” paragraph is not related to the manuscript but mainly reports the strengths and limitations of the four included studies.
Reply: We revised the “Strengths and limitations” section based on the current manuscript as you may see in page 2 of 16, lines 296 – 303.
References
- Habtewold TD, Rodijk LH, Liemburg EJ, Sidorenkov G, Boezen HM, Bruggeman R, et al. A systematic review and narrative synthesis of data-driven studies in schizophrenia symptoms and cognitive deficits. Translational Psychiatry. 2020;10(1). doi: 10.1038/s41398-020-00919-x.
- Hao J, Tiles-Sar N, Liemburg EJ, Habtewold TD, Bruggeman R, van der Meer L, et al. Multidimensional social inclusion and its prediction in schizophrenia spectrum disorder. 2023.
- Quee PJ, Alizadeh BZ, Aleman A, van den Heuvel ER. Cognitive subtypes in non-affected siblings of schizophrenia patients: characteristics and profile congruency with affected family members. Psychol Med. 2014;44(2):395-405. Epub 20130509. doi: 10.1017/s0033291713000809. PubMed PMID: 23659373.
- Quee PJ, Meijer JH, Islam MA, Aleman A, Alizadeh BZ, Meijer CJ, et al. Premorbid adjustment profiles in psychosis and the role of familial factors. Journal of Abnormal Psychology. 2014;123:578-87. doi: 10.1037/a0037189.
- Stiekema APM, Islam MA, Liemburg EJ, Castelein S, van den Heuvel ER, van Weeghel J, et al. Long-term course of negative symptom subdomains and relationship with outcome in patients with a psychotic disorder. Schizophr Res. 2018;193:173-81. Epub 20170622. doi: 10.1016/j.schres.2017.06.024. PubMed PMID: 28648915.
- Tiles-Sar N, Habtewold TD, Liemburg EJ, van der Meer L, Bruggeman R, Alizadeh BZ. Understanding Lifelong Factors and Prediction Models of Social Functioning After Psychosis Onset Using the Large-Scale GROUP Cohort Study. Schizophr Bull. 2023. Epub 20230427. doi: 10.1093/schbul/sbad046. PubMed PMID: 37104875.
- Habtewold TD, Tiles-Sar N, Liemburg EJ, Sandhu AK, Islam MA, Boezen HM, et al. Six-year trajectories and associated factors of positive and negative symptoms in schizophrenia patients, siblings, and controls: The GROUP study. 2023.
- Habtewold TD, Liemburg EJ, Islam MA, De Zwarte SMC, Boezen HM, Bruggeman R, et al. Association of schizophrenia polygenic risk score with data-driven cognitive subtypes: A six-year longitudinal study in patients, siblings and controls. Schizophrenia Research. 2020;223:135-47. doi: 10.1016/j.schres.2020.05.020.
- Islam MA, Habtewold TD, Van Es FD, Quee PJ, Van Den Heuvel ER, Alizadeh BZ, et al. Long-term cognitive trajectories and heterogeneity in patients with schizophrenia and their unaffected siblings. Acta Psychiatrica Scandinavica. 2018;138(6):591-604. doi: 10.1111/acps.12961.
- Habtewold TD, Islam MA, Liemburg EJ, Bruggeman R, Alizadeh BZ, Bartels-Velthuis AAA, et al. Polygenic risk score for schizophrenia was not associated with glycemic level (HbA1c) in patients with non-affective psychosis: Genetic Risk and Outcome of Psychosis (GROUP) cohort study. Journal of Psychosomatic Research. 2020;132:109968. doi: 10.1016/j.jpsychores.2020.109968.
Round 2
Reviewer 1 Report (Previous Reviewer 3)
jpm-2272033-v2: Habtewold et al: Marching towards precision psychiatry
This is a re-review.
The authors compared the results of their review of 53 Schizophrenia Spectrum Disorder (SSD) studies with the results of their longitudinal SSD study of 1,119 patients, 1,059 first-degree relatives, and 586 healthy control subjects, using several different instruments assessing positive, negative, and cognitive symptoms.
They identified 2-6 latent subtypes that were comparable across studies and could be used to model the clinical course in the longitudinal study. They mentioned that the resulting trajectories were predictable by genetic, sociodemographic, social functioning, and disorder-related factors.
And, the authors concluded that (1) baseline positive and negative symptoms, premorbid adjustment, psychotic-like experiences, health-related quality of life and PRSscz were found to be strong predictors of the identified subtypes; and (2) identification of latent subtypes and their clinical courses may help to design randomized control trials and implement targeted evidence-based interventions with maximized treatment efficacy and minimized costs and side effects.
Comments
The authors responded quite well and detailed to the reviewers' questions and comments, and rewrote larger parts of the manuscript. This improved the manuscript considerably, especially its readability.
Nevertheless, the scientific value of the manuscript is still very modest. A clinical value is still not apparent to me, so that it possesses a low publication priority. In consequence, I don’t want to give a recommendation regarding the publication of the manuscript, but would like to leave this decision to the editor.
Author Response
Reviewer 1
This is a re-review.
The authors compared the results of their review of 53 Schizophrenia Spectrum Disorder (SSD) studies with the results of their longitudinal SSD study of 1,119 patients, 1,059 first-degree relatives, and 586 healthy control subjects, using several different instruments assessing positive, negative, and cognitive symptoms.
They identified 2-6 latent subtypes that were comparable across studies and could be used to model the clinical course in the longitudinal study. They mentioned that the resulting trajectories were predictable by genetic, sociodemographic, social functioning, and disorder-related factors.
And, the authors concluded that (1) baseline positive and negative symptoms, premorbid adjustment, psychotic-like experiences, health-related quality of life and PRSscz were found to be strong predictors of the identified subtypes; and (2) identification of latent subtypes and their clinical courses may help to design randomized control trials and implement targeted evidence-based interventions with maximized treatment efficacy and minimized costs and side effects.
Comments
The authors responded quite well and detailed to the reviewers' questions and comments, and rewrote larger parts of the manuscript. This improved the manuscript considerably, especially its readability.
Reply: We thank the reviewer for taking the time to reassess the manuscript and provide your valuable feedback. We appreciate the perspective on the scientific value and the consideration of clinical relevance.
Nevertheless, the scientific value of the manuscript is still very modest. A clinical value is still not apparent to me, so that it possesses a low publication priority. In consequence, I don’t want to give a recommendation regarding the publication of the manuscript, but would like to leave this decision to the editor.
Reply: We would like to clarify that this manuscript aims to succinctly summarize finding from three Ph.D. projects consisting of eight different chapters targeting at deep phenotyping of schizophrenia spectrum disorders (SSD). Its purpose is to present the overarching scientific value of the findings and offer new perspectives that cannot be fully appreciated by individually reading each chapter. This article is not just the summary previously published studies. Rather, we amalgamated main findings, tried to show full picture of SSD phenotypes in patients and healthy individuals, and forwarded future research directions that will help to further the existing scientific knowledge. Therefore, we believe the article have important scientific and clinical values.
Heterogeneity in traditional classification of mental disorders, including schizophrenia spectrum disorders (SSD) is the main challenge in research and clinical practice [1, 2]. As such, key scientific values embodied in this article are highlighted as follows. Firstly, we combined data-driven, polygenic scoring and subphenotyping approaches to evaluate the profiles and longitudinal course of SSD phenotypes by incorporating not only clinical but also psychosocial outcomes in patients and siblings. To our knowledge, there is no cohort that comprehensively investigated positive and negative symptoms, cognitive impairments and social outcomes and compared findings across patients, siblings, and controls. Most previous studies focused on positive and negative symptoms in patients, followed by cognitive impairment [3]. Studies that consistently examined the role of polygenic risk score to predict latent subtypes in positive and negative symptoms are very limited, let alone in social outcomes. Secondly, we depicted the presence of high heterogeneity and tried to show full picture of SSD phenotypes, which is not addressed in previous literature. Thirdly, we conducted a comprehensive data analysis using various complementary statistical modeling techniques to get insight into the prominent predictors of latent subtypes. The combination of using data-driven approaches, and inclusion of PRS and exposomes enhances our understanding of SSD and its trajectory over time. Lastly, the novel findings presented in this article – particularly in siblings’ population and psychosocial functioning phenotypes – can substantially contribute to the existing body of scientific knowledge and open new avenues for future investigations and potential clinical applications.
The findings presented in this article also have also clinical relevancies and will contribute to a broader understanding of the clinical heterogeneity and disease course in SSD in two ways. First, data-driven methods can identify subgroup of SSD patients based on their phenotypes, which may ultimately help refine psychiatric nosology [2]. Additionally, it helps to reduce the heterogeneity for a more targeted assessment of clinical and bio-logical predictors of these phenotypes in SSD. The large number of subtypes observed in studies also alert clinicians not to use generic approaches to treat patients and consider siblings or family members during treatment. Second, the identification of specific biomarkers and clinical factors that are associated with specific latent subtypes will allow for the tailoring of treatments to alleviate positive and negative symptoms and improve cognitive performance and psychosocial functioning in SSD. Identified the predictors that can be used as a marker for clinicians for precisely identifying high-risk groups or patients with similar phenotypes. This could minimize misdiagnosis that commonly seen in schizophrenia [4]. Overall, these findings may have the potential to impact patient care and early preventive and treatment strategies, leading to improved patient outcomes in the long run.
To accommodate this general comment, we added the scientific value (2 of 16, lines 285 - 306) and elaborated the clinical relevance (2 of 16, lines 307 – 321) in the revised manuscript.
We hope we sufficiently addressed your concerns on the scientific values and clinical relevance of our article. Finally, we respect your decision to defer the recommendation on publication to the editor and appreciate your thoughtful assessment of the scientific value and consideration of the clinical implications.
References
- Allsopp K, Read J, Corcoran R, Kinderman P. Heterogeneity in psychiatric diagnostic classification. Psychiatry Research. 2019;279:15-22. doi: 10.1016/j.psychres.2019.07.005.
- Feczko E, Miranda-Dominguez O, Marr M, Graham AM, Nigg JT, Fair DA. The Heterogeneity Problem: Approaches to Identify Psychiatric Subtypes. Trends in Cognitive Sciences. 2019;23(7):584-601. doi: https://doi.org/10.1016/j.tics.2019.03.009.
- Habtewold TD, Rodijk LH, Liemburg EJ, Sidorenkov G, Boezen HM, Bruggeman R, et al. A systematic review and narrative synthesis of data-driven studies in schizophrenia symptoms and cognitive deficits. Translational Psychiatry. 2020;10(1). doi: 10.1038/s41398-020-00919-x.
- Lopez-Castroman J, Leiva-Murillo JM, Cegla-Schvartzman F, Blasco-Fontecilla H, Garcia-Nieto R, Artes-Rodriguez A, et al. Onset of schizophrenia diagnoses in a large clinical cohort. Scientific Reports. 2019;9(1). doi: 10.1038/s41598-019-46109-8.
Reviewer 6 Report (New Reviewer)
Considering the authors' claim to provide an overview of findings from the Genetic Risk and Outcome of Psychosis (GROUP) cohort studies on deep clinical phenotyping of schizophrenia spectrum disorders, their article falls short of meeting the requirements expected of a perspective article. A perspective article should involve a critical evaluation of relevant literature, identification of key studies, theories, and concepts, and the presentation of a balanced overview of existing knowledge, controversies, and gaps in understanding. However, the authors' exclusive focus on the GROUP cohort studies prevents them from offering a comprehensive analysis of the broader evidence reported by other studies.
Furthermore, the authors make criticisms toward the Diagnostic and Statistical Manual of Mental Disorders (DSM) without providing adequate evidence to support their claims. For example, their reference to a study suggesting a 50% misdiagnosis rate (13) is misleading, as the study specifically pertains to schizophrenia and not to DSM-5 diagnoses in general. Additionally, their assertion that the DSM-5 renamed schizophrenia to schizophrenia spectrum disorder (SSD) is factually incorrect. In reality, the DSM-5 introduced the concept of schizophrenia spectrum disorders, encompassing various diagnoses, including schizophrenia.
These substantial concerns and the absence of a comprehensive and balanced analysis lead to the recommendation of not suggesting the publication of the article. The authors should revise their work to address these deficiencies and provide a more robust and unbiased assessment of the relevant literature, as well as the implications of the DSM-5 concerning schizophrenia and schizophrenia spectrum disorders.
Furthermore, it is worth noting that the authors included a citation from an article that was still under review (19). Consequently, this citation cannot be considered as scientifically valid evidence since the article has not undergone a rigorous peer-review process or received acceptance for publication. In a perspective article, it is crucial to rely on established, peer-reviewed literature to support claims and arguments. Additionally, another citation (24) lacks sufficient bibliography, which raises further concerns.
Taking into account these additional concerns, in conjunction with the aforementioned shortcomings in the article, the manuscript cannot be recommended for publication.
Author Response
Reviewer 6
We thank the reviewer for taking the time again to reassess the manuscript and provide his/her valuable feedback. We appreciate the perspective on the type of article, prospective studies and the consideration of appropriate referencing.
Concern 1: Considering the authors' claim to provide an overview of findings from the Genetic Risk and Outcome of Psychosis (GROUP) cohort studies on deep clinical phenotyping of schizophrenia spectrum disorders, their article falls short of meeting the requirements expected of a perspective article. A perspective article should involve a critical evaluation of relevant literature, identification of key studies, theories, and concepts, and the presentation of a balanced overview of existing knowledge, controversies, and gaps in understanding. However, the authors' exclusive focus on the GROUP cohort studies prevents them from offering a comprehensive analysis of the broader evidence reported by other studies.
Reply: Indeed, we agree that a perspective article should provide a critical evaluation of relevant literature, identification of key studies, theories, and concepts, and present a balanced overview of existing knowledge, controversies, and gaps in understanding. While this manuscript aims to succinctly summarize three Ph.D. projects consisting of eight different chapters, based on studies conducted within the framework of the large-scale longitudinal GROUP cohort on deep clinical phenotyping of schizophrenia, we have recently published (1) a highly comprehensive systematic review of all relevant literature ensuring a balanced overview of existing knowledge and with detailed discussion over controversies that was necessary to meet the requirements of a perspective article. This comprehensive review expanded the scope and incorporated insights from other studies to provide a more comprehensive perspective on the topic. We have cited this full-spectrum review as reference 30, and summarized its main findings in the discussion section.
We believe, therefore, it is beyond the scope and aim of this manuscript to perform yet another full review, while the existing recent one covers the scope and opinion of this reviewer and the field adequately. Therefore, after discussion with the editor and co-authors, we requested to change the article type to “Brief Report” during resubmission of the revised manuscript.
Concern 2: Furthermore, the authors make criticisms toward the Diagnostic and Statistical Manual of Mental Disorders (DSM) without providing adequate evidence to support their claims. For example, their reference to a study suggesting a 50% misdiagnosis rate (13) is misleading, as the study specifically pertains to schizophrenia and not to DSM-5 diagnoses in general. Additionally, their assertion that the DSM-5 renamed schizophrenia to schizophrenia spectrum disorder (SSD) is factually incorrect. In reality, the DSM-5 introduced the concept of schizophrenia spectrum disorders, encompassing various diagnoses, including schizophrenia.
Reply: Thank you for bringing these concerns to our attention. We acknowledge the need for adequate evidence to support any criticisms or claims made in the manuscript. Regarding the reference to a study suggesting a 50% misdiagnosis rate, we understand that the study specifically pertains to schizophrenia and not to DSM-5 diagnoses in general. Our evidence was also referred to schizophrenia misdiagnose rate, not referring to other mental disorders listed in DSM. To minimize unnecessary confusion, we revised the sentence as you may see in page 2 of 16, lines 71 – 72. Moreover, we appreciate the reviewer clarification on DSM-5 concept of schizophrenia spectrum disorders. Therefore, we revised it accordingly on page 2 of 16, lines 84 - 85 to minimize misunderstanding.
Concern 3: These substantial concerns and the absence of a comprehensive and balanced analysis lead to the recommendation of not suggesting the publication of the article. The authors should revise their work to address these deficiencies and provide a more robust and unbiased assessment of the relevant literature, as well as the implications of the DSM-5 concerning schizophrenia and schizophrenia spectrum disorders.
Reply: Please see our replies for concerns 1 and 2. We hope we sufficiently addressed your concerns.
Concern 4: Furthermore, it is worth noting that the authors included a citation from an article that was still under review (19). Consequently, this citation cannot be considered as scientifically valid evidence since the article has not undergone a rigorous peer-review process or received acceptance for publication. In a perspective article, it is crucial to rely on established, peer-reviewed literature to support claims and arguments. Additionally, another citation (24) lacks sufficient bibliography, which raises further concerns.
Reply: Indeed, we agree that it is crucial to rely on established, peer-reviewed literature to support claims and arguments. As mentioned before, we requested to change the article type to “Brief Report” during resubmission of the revised manuscript. Reference #19 has been revised and resubmitted for the 2nd time, and is currently awaiting a decision by the Editor. Secondly, we have recently published another in-depth study partly using findings (subtypes of positive and negative symptoms) from reference #19. We have included this study as reference 26 (2). Moreover, we included the URL for citation reference #24 (https://doi.org/10.21203/rs.3.rs-2608209/v1).
Concern 5: Taking into account these additional concerns, in conjunction with the aforementioned shortcomings in the article, the manuscript cannot be recommended for publication.
Reply: Please see our replies for concerns 1, 2, and 4. We hope we sufficiently addressed your concerns. We would also like to note that this article is not only the summary of previously published studies. Rather, we amalgamated main findings, tried to show full picture of SSD phenotypes in patients and healthy individuals, and forwarded future research directions that will help to further the existing scientific knowledge. Therefore, we believe the article have important scientific and clinical values when published.
Once again, we thank you for your time and effort in reviewing the manuscript. Your input is highly valued, and we were committed to addressing these concerns and improving the manuscript accordingly.
References
- Habtewold TD, Rodijk LH, Liemburg EJ, Sidorenkov G, Boezen HM, Bruggeman R, et al. A systematic review and narrative synthesis of data-driven studies in schizophrenia symptoms and cognitive deficits. Translational Psychiatry. 2020;10(1).
- Tiles-Sar N, Habtewold TD, Liemburg EJ, van der Meer L, Bruggeman R, Alizadeh BZ. Understanding Lifelong Factors and Prediction Models of Social Functioning After Psychosis Onset Using the Large-Scale GROUP Cohort Study. Schizophr Bull. 2023.
This manuscript is a resubmission of an earlier submission. The following is a list of the peer review reports and author responses from that submission.
Round 1
Reviewer 1 Report
Main point
The main point is that the manuscript seems to be just a summary of previous studies published by the research group. The unsuspecting reader will expect other content under the presented title of the manuscript. That said, other considerations would be unnecessary. However, here are some considerations:
Other major points
§ The authors classify the manuscript as a “brief report”. I believe that it may be brief when thinking about post-doctoral production, but it is extensive when thinking about a scientific journal manuscript described under this term;
§ The GBTM is usually described as allowing a simple and easy-to-understand representation - which would not exempt the results (as Figures 2 and 3) data from being better described and discussed. The subgroups (subphenotypes) must be better explained and their constructs better proposed;
§ The authors must consider the inclusion of important information for the evaluation of the internal and external validity of what they defend as results (and without the reader depending on reading the group's previous publications) - such as adherence to treatment (as a possible confounder evolution of groups) and something about the PRS;
§ Figures and tables without enough information to make them independent objects of the text [to mention a few: lack of standardization in the presentation (for example, the name of Figure 1 is all in bold); lack of presentation of the acronyms used or the explanation of the acronym comes after its first use; the reader must assume that the numbers in parentheses are actually the references of studies already published; absence of legends for conditions (a), (b), (c) and (d) of Figure 3]. And, although manuscripts published by JPM are not restricted in length, Figure 1 does not appear to add additional information essential for the reader's understanding.
Some minor points
§ To make it easier for the reader, I suggest the authors to review the order in which they describe the methodology of analysis. It would be better to be done in the same order as the studies (from 1 to 5). Thus, the mention of what was done in study 1 (which, by the way, it was not a statistical analysis) could be described earlier (and not at the end of the session);
§ There is a text highlight for Department information (line 8) and it was not possible to understand what it was about;
§ There are points of formatting errors. It would be necessary to remove hyphenation where unnecessary (for example, “patients” on line 100). Also, there was doubt about the appropriateness of using parentheses in phrases such as “treat using (non)pharmacological interventions”. Does this mean that the authors are referring to both pharmacological and non-pharmacological interventions? If so, it may not seem so intuitive to the reader;
§ Likewise, I believe that there was an error in generating the PDF, as the sentence that starts on line 169 must have been unformatted and lost its meaning (see “Finally, in Chapter III, chemical dysregulation in patients. Details about the study results are summarized in Table 2. 170 (Study 5), we found that late age at first psychosis is significantly associated with gly”);
§ The use of a correspondence between chapter/study does not seem to be necessary.
Author Response
Reviewer #1
We value the commitment, time and the effort given to review our manuscript. We are very grateful for allowing us to revise and improve our manuscript. Hereunder, we tried to address your concerns mainly about the subgroups, validation, PRS, and editorials.
- The authors classify the manuscript as a “brief report”. I believe that it may be brief when thinking about post-doctoral production, but it is extensive when thinking about a scientific journal manuscript described under this term.
Reply: Thank you for appreciating the extensive nature of our work. We will consider publishing as a regular article after discussing with the editor.
2. The GBTM is usually described as allowing a simple and easy-to-understand representation - which would not exempt the results (as Figures 2 and 3) data from being better described and discussed. The subgroups (subphenotypes) must be better explained and their constructs better proposed.
Reply: Thank you for this suggestion. We now provide detailed explanation of the subgroups in Table 3 and included in the revised manuscript as well (see Table 3, page 11 of 18). Now we hope the difference between subgroups is clear.
3. The authors must consider the inclusion of important information for the evaluation of the internal and external validity of what they defend as results (and without the reader depending on reading the group's previous publications) - such as adherence to treatment (as a possible confounder evolution of groups) and something about the PRS.
Reply: We totally agree on the importance of validating the identified subgroups and the role of PRS. The subgroups and role of PRS to distinguish these subgroups were not externally validated in the original studies, which could be future endeavors and put it as a limitation in our studies (page 15 of 18, lines 346 - 348). The adherence to treatment can also contribute to the difference in latent classes and change of symptomatic trajectories. However, because of lack of data in our cohort, we could not be able to check the role of patients’ adherence to treatment, which requires further investigation. We included this comment as future research perspective and our research groups are also working by extending the original studies (see page 14 of 18, lines 298 - 299).
- Figures and tables without enough information to make them independent objects of the text [to mention a few: lack of standardization in the presentation (for example, the name of Figure 1 is all in bold); lack of presentation of the acronyms used or the explanation of the acronym comes after its first use; the reader must assume that the numbers in parentheses are actually the references of studies already published; absence of legends for conditions (a), (b), (c) and (d) of Figure 3]. And, although manuscripts published by JPM are not restricted in length, Figure 1 does not appear to add additional information essential for the reader's understanding.
Reply: This is important for the clarity and readability of the manuscript. Throughout the revised manuscript, we uniformly presented figure and table titles, explained acronyms at first use in the text and properly described when used in the tables, and legends are added for all figures. Finally, we deleted Figure 1 as recommended.
- To make it easier for the reader, I suggest the authors to review the order in which they describe the methodology of analysis. It would be better to be done in the same order as the studies (from 1 to 5). Thus, the mention of what was done in study 1 (which, by the way, it was not a statistical analysis) could be described earlier (and not at the end of the session).
Reply: In the revised manuscript, we included a separate section (Section 2.3) describing the methodology of Study 1 (see page 5 of 18, lines 150 – 160)
- There is a text highlight for Department information (line 8) and it was not possible to understand what it was about.
Reply: The text highlight for ‘Department’ information was automatically done by the journal submission system. We submitted clean/unhighlighted text (see page 1, lines 8 - 15). We hope it will not again be highlighted.
- There are points of formatting errors. It would be necessary to remove hyphenation where unnecessary (for example, “patients” on line 100). Also, there was doubt about the appropriateness of using parentheses in phrases such as “treat using (non)pharmacological interventions”. Does this mean that the authors are referring to both pharmacological and non-pharmacological interventions? If so, it may not seem so intuitive to the reader.
Reply: In the revised manuscript, we double checked and corrected all the possible formatting errors. Additionally, we removed using parentheses in phrases as you indicated and wrote in full to minimize confusion.
- Likewise, I believe that there was an error in generating the PDF, as the sentence that starts on line 169 must have been unformatted and lost its meaning (see “Finally, in Chapter III, chemical dysregulation in patients. Details about the study results are summarized in Table 2. 170 (Study 5), we found that late age at first psychosis is significantly associated with gly”).
Reply: We corrected this mistake in the revised manuscript (see page 7 of 18, lines –194 -196).
Table 3. Description of positive, negative and cognitive symptoms trajectories
|
Positive symptoms |
Negative symptoms |
Cognitive symptoms |
Patients |
3 trajectories (Figure 1A) |
3 trajectories (Figure 1D) |
5 trajectories (Figure 2A) |
Unaffected siblings |
4 trajectories (Figure 1B) |
4 trajectories (Figure 1E) |
4 trajectories (Figure 2B) |
Healthy controls |
3 trajectories (Figure 1C) |
3 trajectories (Figure 1F) |
4 trajectories (Figure 2C) |
All samples |
NA |
NA |
6 trajectories (Figure 2D) |
Abbreviations: PANSS: Positive and Negative Syndrome Scale; SIS-R: Structured Interview for Schizotypy-Revised; SD: standard deviation
References:
- Tesfa D. Habtewold EJL, Richard Bruggeman, Behrooz Z. Alizadeh. Symptomatic trajectories and clusters in patients with schizophrenia, siblings and healthy controls PROSPERO: PROSPERO; 2018 [Available from: https://www.crd.york.ac.uk/prospero/display_record.php?ID=CRD42018093566.
- Habtewold TD, Rodijk LH, Liemburg EJ, Sidorenkov G, Boezen HM, Bruggeman R, et al. A systematic review and narrative synthesis of data-driven studies in schizophrenia symptoms and cognitive deficits. Translational Psychiatry. 2020;10(1).
- Roffman JL. Endophenotype Research in Psychiatry-The Grasshopper Grows Up. JAMA Psychiatry. 2019;76(12):1230-1.
- Donati FL, D’Agostino A, Ferrarelli F. Neurocognitive and neurophysiological endophenotypes in schizophrenia: An overview. Biomarkers in Neuropsychiatry. 2020;3:100017.
- McElreath R. Statistical rethinking: A Bayesian course with examples in R and Stan: Chapman and Hall/CRC; 2020.
- Murphy KP. Probabilistic machine learning: an introduction: MIT press; 2022.

Reviewer 2 Report
Overall, my opinion is that this manuscript is interesting and well-written. There is some data on this important topic. Generally, the paper has high issues with the standard of writing.
The manuscript could be strengthened by attending to the following matters:
1) GENERAL COMMENTS
Positive:
- Interesting topic
Negative:
- Some limitations (methodology)
Abstract:
Abstract:
Please replace »we« with »authors..«
INTRODUCTION
Lines 65-71, page 2/16: Schizophrenia is especially comorbid in elderly patients, which minimizes treatment guidelines, often not focused on »real patients« but more on RCTs. In this context, more »real« trials are needed in diagnosing and treating elderly patients: »Antipsychotic treatment in elderly patients on polypharmacy with schizophrenia. Curr Opin Psychiatry. 2022 Sep 1;35(5):332-337. doi: 10.1097/YCO.0000000000000808.«
MATERIAL AND METHODS
2. Materials and Methods
2.1. Study population
Study 1: Did the authors use a systematic review method? They should better describe their methodology.
2.2. Measured variables
Please specify primary and secondary outcomes.
2.3. Statistical analysis
Did the authors also check data quality (e.g., GRADE)? Please determine if specific protocols were used (e.g. STROBE).
Results
Figure 4: Very hard to read. Please improve.
Discussion
Please include more limitations (especially study design and bias).
Author Response
Reviewer #2
Thank you very much for the appreciation of our work, and your time and the effort given to critically review our manuscript. Hereunder, we tried to address comments.
1. Please replace »we« with »authors.
Reply: We replaced ‘we’ with ‘authors’ throughout the revised manuscript.
- Lines 65-71, page 2/16: Schizophrenia is especially comorbid in elderly patients, which minimizes treatment guidelines, often not focused on »real patients« but more on RCTs. In this context, more »real« trials are needed in diagnosing and treating elderly patients: »Antipsychotic treatment in elderly patients on polypharmacy with schizophrenia. Curr Opin Psychiatry. 2022 Sep 1;35(5):332-337. doi: 10.1097/YCO.0000000000000808.«
Reply: We included this text in the revised manuscript and the reference was properly cited (see page 2 of 18, lines 70 - 72).
- Study 1: Did the authors use a systematic review method? They should better describe their methodology.
Reply: In study 1, we used systematic review method following PRISMA guideline and the authors’ registered protocol (see page 3 of 18, line –100; page 5 of 18, lines 150 - 160).
- Please specify primary and secondary outcomes.
Reply: We specified the primary and secondary outcomes in the revised manuscript (see page 5 of 18, lines 127 and 136).
5. Did the authors also check data quality (e.g., GRADE)? Please determine if specific protocols were used (e.g. STROBE).
Reply: Data quality check is indeed of a vital importance. Study 1 was a systematic review conducted following the Preferred Reporting Items for Systematic Review and Meta-Analysis (PRISMA-protocol) statement and authors’ pre-registered protocol to ensure the standard procedure and quality of the review (page 5 of 18, lines 150-160). The strength of evidence was not measured. We included this as a limitation (see page 15 of 18, line 346 - 348).
6. Figure 4: Very hard to read. Please improve.
Reply: All the figures in the manuscript are not clearly readable because of a lot of compression to fit in the word file for submission. We will provide the original, publication standard figure (.tiff file) up on acceptance of our manuscript.
- Please include more limitations (especially study design and bias).
Reply: We included additional limitations in the revised manuscript (page 15 of 18, line 346 - 348). We also forward our recommendation to address these limitations in future studies (page 14 of 18, lines 267 -317).
References:
- McElreath R. Statistical rethinking: A Bayesian course with examples in R and Stan: Chapman and Hall/CRC; 2020.
- Murphy KP. Probabilistic machine learning: an introduction: MIT press; 2022.

Reviewer 3 Report
JPM-1858503: Marching towards precision psychiatry
In this PhD summary report, the results of previously published studies were used as a basis for attempts to resolve the phenotypic heterogeneity of schizophrenia spectrum disorders by means of data-driven methods. The data used were: (1) positive and negative symptoms assessed by PANSS (patients) or SIS-R (siblings and healthy controls); (2) cognitive functioning assessed by Word Learning Task (WLT), Continuous Performance Test (CPT-HQ), and Wechsler Adult Intelligence Scale (WAIS-III); (3) sociodemographic variables; and (4) various clinical and functional variables. The data analysis included descriptive statistics, chi-square tests, linear regression, and group-based trajectory modelling.
The authors found a continuum of positive, negative, and cognitive symptoms across patients, siblings, and health controls. Furthermore, they report that they identified 2-6 subtypes of positive, negative, and cognitive symptoms that could be distinguished by genetic susceptibility and other variables. And they conclude that the use of data-driven approaches may be useful for diagnostic purposes and treatment selection.
Comment
This is certainly a carefully elaborated PhD thesis whose clinical value, however, is difficult to assess. Especially because its potential clinical application regarding diagnostic purposes or the selection of improved treatment options is insufficiently outlined.
The question also arises as to why more advanced methods such as AI approaches have not been taken into consideration. This should be discussed.
Author Response
Reviewer #3
Thank you very much for your appreciation, and time and the effort given to review our manuscript. We are very grateful for allowing us to revise and improve our manuscript. Hereunder, we tried to address your concerns in terms of clinical application and other advanced AI approaches.
1. This is certainly a carefully elaborated PhD thesis whose clinical value, however, is difficult to assess. Especially because its potential clinical application regarding diagnostic purposes or the selection of improved treatment options is insufficiently outlined.
Reply: We elaborated the clinical implication of the PhD project (see page 15 of 18, lines 323 - 335).
2. The question also arises as to why more advanced methods such as AI approaches have not been taken into consideration. This should be discussed.
Authors’ reply: It is true that the rapid advancement of AI approaches has been providing substantial opportunities to explore high-volume high-dimensional data. In the revised manuscript, we explained the benefit of applying AI approaches (machine learning perspective) and forward our recommendation for future researchers (see page 14 of 18, lines 303 – 317)

Reviewer 4 Report
Thank you for giving me this opportunity to review this paper. I understand the project has been a huge one with different stages, although the report by the paper is complicated, indeed. The rationale, methods and results are reported confusing and I believe that combination of the report of different stages made it hard to understand. All in all, I don`t think that it`s suitable for the publication in the current form.
Some additional comments:
1. 'of the four years PhD project' is not useful in the abstract.
2. In my opinion, the abstract is not coherent enough and I don`t get the methods and findings of the study after reading it.
3. This sentence is not useful enough: 'This PhD summary article is written based on the PhD thesis submitted and defended at the department of Epidemiology, University Medical Center Groningen (UMCG), University of Groningen on February 02, 2021.'
Reviewer 5 Report
In line 202, Numbers in parenthesis --> Numbers in parentheses
In Conclusion, lines 350-353, Through this PhD summary report, in a combinative application of the data-driven, PRS and subphenotyping approaches, the authors’ found that SSD constitutes up to six latent subgroups of symptoms that are characterized by stable, increasing, decreasing and relapsing trajectories over time. --> A combinative application of the data-driven, PRS and subphenotyping approaches showed that SSD constitutes up to six latent subgroups of symptoms characterized by stable, increasing, decreasing and relapsing trajectories over time.
The rest is OK
Reviewer 6 Report
This manuscript (JPM-1858503) titled Moving towards precision psychiatry: Data-driven subphenotypic dissection of heterogeneity in schizophrenia spectrum disorders represents a summary of previous published research from the Genetic Risk and Outcome of Psychosis (GROUP) longitudinal study, as well as a review study. The manuscript is based on a larger (more in-depth) PhD project where information is organized by chapters (depicted clearly in Table 1).
While I appreciate the intent of this project, I found it quite difficult to piece everything together while reading. This is probably not an issue in the larger document where more information is likely provided. I'm not sure if the Brief Report format may be hindering clarity. I was constantly trying to gain context to details that are only provided in other source materials, or presented in a fragmented way in different tables. Therefore I had a very difficult time assessing the statements/claims being made in the current document. This sense of disorientation was acute when attempting to make sense of the Figure 1 charts. The figure is referenced in a later Table (Table 3), but it’s still not clear what publication this data is extracted from. I can clearly see that the lines represent symptom severity scores that are stratified overtime, but I have to context to judge the significance of these stratifications, or how the occasional “high-decreased” or “high-increased” groups were identified. I believe this might be one of the papers still “under review”. I find this a little odd since the source reference paper is not available (and may never be available, or may change during the peer review process). Similarly, there are references to polygenic risk scores. However, very little information is provided about what these scores represent. In section 2.4 the authors state that polygentic risk scores were “summarized using frequency and means”, but this statement does not state what those frequencies or means represent. So, unless I look up the reference for study 4 (Habtewold et al., 2020), I will not really have a context to interpret the meaning of PRS profiles in Figure 3. That is, I can see that PRSp=0.5 in the patient group has mild-severe cognitive profiles, but that has little meaning without knowing what PRS represents within the current paper. I guess this makes me wonder who you intend to read this paper. It might be perfectly suited for a small group of researchers who are intimately familiar with the research of your group (who would presumably already know the info you are presenting) but it will have little meaning/relevance to those with a more general interest in learning about “marching towards precision psychiatry”.
I was a little concerned by the characterization of the DSM (which made me question the author’s understanding of how diagnoses are made using this document). The DSM has not “shifted away from the categorical diagnostics and adopted a dimensional approach”. It is very much still a categorical diagnostic system. There was a “name only” change where the DSM added “spectrum” to the title of the diagnostic section. Schizotypal personality disorder and the other cluster A disorders are still in a later section (and also categorical). Also, the authors state that diagnoses are “based solely on the DSM criteria, where clinicians subjectively interview patients using neuropsychological assessment instruments”. This statement has me confused since no “neuropsychological assessment instruments” are used in the process of diagnosing.
I appreciated the future perspectives and suggestions for additional research directions. Although the “precision psychiatry” element (also suggested in the title) seems to be extrapolating far from the current findings. It’s not clear how this specific data-driven strategy differs from other research approaches, or why this research will contribute to targeted psychiatric interventions. Section 6. Steps towards dynamic precision psychiatry and clinical implications would benefit from an explicit statement of how their collective findings lead to a specific recommendation. They suggest that data-driven methods, such as the ones presented in the paper, could “assist in the implementation of personalized and preventative strategies for clinical practice at least in the national or regional level”. They provide a suggestion that this approach could be used in pharmacological vs. non-pharmacological research. But I am not convinced that this would lead to precision medicine (especially given the heterogeneity they rightfully point out).
Round 2
Reviewer 1 Report
Unfortunately, I maintain my consideration of the main point of the study - I get the impression that it is a summary of previous productions, without adding an original proposal, as would be suggested by the title (including the use of terms with "precision" and "dissection").
I apologize because I see that I was not clear when asking for the data to be better explained. There is no need to include Table 3 – the information included in table 3 can be turned into a legend for the figures (although I imagine it will be a long legend).
I understood that the authors defend that cases of non-affective psychosis should be better described in dimensional terms - and, by the title, they intend to dissect the observed heterogeneity in more intended homogeneous subgroups.
First, the empirically proposed profiles (with a sample of an already studied cohort) “do not talk to” the findings described in what the authors call (other) chapters.
Moreover, the description and consideration of the results allow the conceptualization of which profiles and which predictions? For example, from the graphs, it would be possible to suggest that the evolution of SSD does not show progression, at least in 6 years of follow-up. Patients which positive symptom classified as “moderate” remain moderate? Do those with symptoms considered high show improvement in 3 years, but almost return to study entry grade and show no significant improvement? Bearing in mind that the authors mention that there are no data that could be used to control bias as to whether or not they are on adequate drug treatment, what did it mean?
Another example, although we still have nowadays little success in treating negative symptoms, there is a group that has a decrease in these symptoms in 6 years? So, it meant that they got better and who were this subgroup?
And, what is fundamental, who are these people who composed all these proposed subgroups? In terms of diagnoses classified by the GROUP as SSD - by reference 13: there were, at least, 20% classified as “psychosis NOS” or "others". At least, 50% of the GROUP sample seemed to have had a measure suggestive of substance abuse. So: who are these volunteers in all these proposed profiles?
There is a sibling subgroup that seems to have a greater presentation of negative elements in schizotypy in year 6 - are they a group at greater risk for, if exposed to other risk factors, developing psychosis? Are they the biological siblings of those in the GROUP study who have had the explicit diagnosis of schizophrenia? What does it mean for the group of siblings and controls to have a different number of apparent evolution profiles?
How does this improve the traditional classification into nosological entities? What are the clinical implications of this? Likewise, does the evolution of the cognitive score show that, for the majority, there is no progression of the loss?
Again, because the studies “don't talk to” each other, it gives me the impression of an article summarizing studies that have already been published.
Because the study did not present information and did not investigated associations between the proposed evolutionary profiles and sociodemographic and clinical characteristics of the sample, I maintain that it is not possible to consider both the internal and external validity as sufficient.
Consider que sentence “Furthermore, the trajectories of cognitive performance can be predicted by baseline symptomatic severity”. Could this association not actually be the opposite (if I have a vulnerability in terms of developing cognitive skills and this increases my vulnerability to developing psychosis)? It will then be, with the direction of this association and the data that show a low change in cognitive scores over time, that symptoms imply a loss at the beginning of psychosis, but this impairment is not progressive - because at 3 and 6 years, the profiles are stable? What does this imply, therefore, in terms of investment in treatment?
Then, if the text were to be completed with tables, it would be more interesting for the readers to present the estimates of the regressions apparently made (unless they are just the reproduction of studies already published).
Finally, the acronym “RCTs” was used only once without presenting its meaning.